

# Lake Surface Temperature Dataset in the North Slave Region Retrieved from Landsat Satellite Series - 1984 to 2021

Gifty Attiah [1,2], Homa Kheyrollah Pour [1,2], K. Andrea Scott [3]

[1]Remote Sensing of Environmental Change (ReSEC) Research group, Department of Geography and Environmental Studies, Wilfrid Laurier University, 75 University Ave, West N2L 3C5, Canada
[2]Cold Regions Research Centre, Wilfrid Laurier University, Waterloo, ON, Canada
[3] Department of Systems Design Engineering, University of Waterloo, 200 University Ave, West N2L 3G1, Canada

*Correspondence to*: Gifty Attiah (gattiah@wlu.ca)

**Abstract.** Lake surface temperature (LST) is an important attribute that highlights regional weather and climate variability and trends. The spatial resolution and thermal sensors on Landsat platforms provide the capability of monitoring the temporal and spatial distribution of lake surface temperature on small to medium size lakes. In this study, a retrieval algorithm was applied to the thermal bands of Landsat archives to generate a LST dataset (North Slave LST dataset) for 535 lakes in the North Slave Region (NSR) of the Northwest Territories (NWT), Canada for the period of 1984 to 2021. North Slave LST was retrieved from Landsat-5 TM, Landsat-7 ETM+ and Landsat-8 OLI/TIRS, however majority of the dataset were created from the thermal bands of Landsat-5 (43%) due to its longevity (1984-2013). Cloud masks were applied to Landsat images to eliminate cloud cover. In addition, a 100-meter inward buffer was applied to lakes to prevent pixel mixing with shorelines. To evaluate the algorithm applied, retrieved LST was compared with in-situ data and Moderate Resolution Imaging Spectroradiometer (MODIS) LST observations. A good agreement was observed between in-situ observations and North Slave LST derived in this study with a mean bias of 0.12 °C and an RMSD of 1.7 °C. The North Slave LST dataset contains more available data from warmer months (May to September), covering 57.3 % in comparison to colder months (October to April). Average number of images per year for each lake across the NSR ranged from 20 to 45. The North Slave LST dataset will provide communities, scientists and stakeholders with spatial and temporal changing trends of temperature on lakes for the past 38 years.





## 1 Introduction

Lakes surface temperature (LST) is a significant indicator of climate change and crucial to lake ecosystems (Livingstone et al., 2005; G. Zhang et al., 2019). Several ecological, biological, and hydrogeochemical processes are influenced by temperature in lakes (Schneider & Hook, 2010). Lake warming can result in decrease in ice cover, changes in over lake wind speeds, and changes in water column stratification (Austin & Colman, 2007; Desai et al., 2009; Kraemer et al., 2015; Magnuson et al., 2000). Energy and material exchange processes of land-water-atmosphere system can also be reflected in lake surface temperature (Huang et al., 2017; Yang et al., 2020) and hence recognized as an essential climate variable. As a significant



variable in regional studies, the impact and relationship of LST to weather, climate and lake processes have been explored by other studies including influences on weather (Kheyrollah Pour et al., 2014a,b; Eerola et al, 2014; Kheyrollah Pour et al., 2017), climate (Moigne et al., 2016; Wang et al., 2021), precipitation (Zhang et al., 2016), lake effect snow (Shi & Xie, 2019)

and lake overturning (Fichot et al., 2019). Observations of lakes around the world have reported increases in lake temperature associated with global warming resulting in changes to the underlying lake system (O'Reilly et al., 2015; Woolway et al., 2019). Long-term records of lake surface temperature are therefore necessary to understand thermal mechanism underlying lake processes including lake ice formation and decay, lake productivity, aquatic ecosystems and other limnological processes (Chen et al., 2019; Collingsworth et al., 2017; Woolway et al., 2020).

Even though in-situ records on lake surface temperatures are a good source of temperature data for lakes studies, their sparse distribution especially in the north present a challenge making satellite-derived data an important resource in regional and global studies. Satellite sensors like MODIS (Moderate Resolution Imaging Spectroradiometer) and AVHRR (Advanced Very High Resolution Radiometer) have been heavily relied upon to estimate and analyse LST in several studies (*e.g.,* Kheyrollah Pour et al., 2012, 2014a, b, 2017; Reinart & Reinhold, 2008; Sima et al., 2013; Wan et al., 2002; Wloczyk et al., 2006; Zhao

et al., 2020), however, their application to small and medium lakes is limited due to relatively moderate spatial resolution (~500 m - 1 km). In addition, satellite retrieved LST datasets for global studies like the Global Lake Temperature Collaboration (GLTC) have low sampling of high latitude lakes which restricts their use for climate studies in these northern regions. Satellites like Landsat however provides an opportunity for regional studies of lake processes and spatial extraction of LST including Arctic and subarctic lakes. The strength of Landsat includes its high spatial resolution (30 m -120 m), high

radiometric resolution (8-12 bits) and the presence of thermal infrared bands for the retrieval of LST. In addition, longevity of data archives makes it one of the most extensive and longest observation of earth's surface water from space (Pekel et al., 2016). Currently, a regional spatial lake surface temperature dataset for small and medium size lakes on a large scale does not exist in NWT, more specifically the North Slave Region (NSR) lakes and this study seeks to bridge this gap by using the capabilities of Landsat to achieve this.

In this study we generated LST data (North Slave LST) for over 535 predominantly small to medium lakes using data obtained from Landsat archives (Landsat-5 TM, Landsat-7 ETM+ and Landsat-8 OLI/TIRS). An adapted temperature retrieval algorithm (Jimenez-Munoz et al., 2009, 2014) is applied to the thermal bands of Landsat to estimate LST. The dataset produced has a spatial resolution of 30 m and varying temporal resolution due to differences in satellite overpass and cloud interference. The generated North Slave LST dataset was evaluated with in-situ datasets and compared with widely used LST satellite

dataset (MODIS). Temporal and spatial distribution of the dataset is presented to report on data availability patterns. Additionally, North Slave LST dataset is used to briefly highlight the spatial inter-lake and intra-lake distribution of LST in the NSR lakes.

The aim of this study is to (i) capitalize on the thermal bands of Landsat to create an up-to-date lake surface temperature dataset in the NSR to record distribution from 1984 to 2021; (ii) highlight the temporal and spatial heterogeneity of LST between and





within lakes on a 30 m grid; (iii) Distribute and publish LST data for stakeholders, research communities to facilitate further research and studies, the public and the Government of the Northwest Territories to facilitate decision making processes.

## 2 Study Lakes and Data Sources

### 2.1 Selected Lakes in North Slave Region, NWT

The North Slave LST data is generated for 535 lakes, between latitude 61°N and 67°N and longitude -120°W and -102W of
the Northwest Territories (NWT) located in the northern part of Canada covering an area of about 316,000 km$^2$. The region lies in the Slave province of the Canadian shield and interspersed with numerous lakes (>10,000) in various sizes. Elevation in the NSR has an average altitude of 301 m with lake elevation ranging from 138 m to 624 m (Messager et al., 2016). This dataset contains 535 lakes with surface area ranging from 0.05 km$^2$ to 1680 km$^2$ and mean depths ranging from 1 m – 63 m with volume ranging from 0.24 km$^3$ to 27321 km$^3$. Appendix A contains a list of lakes with geophysical properties. Air
temperature in the NSR ranges from ~-45°C to +30°C. The majority of the study lakes are between an area of 1 and 5 km$^2$ (Figure b) and the dominant mean depth range was 5 – 10 m (Figure 1c).

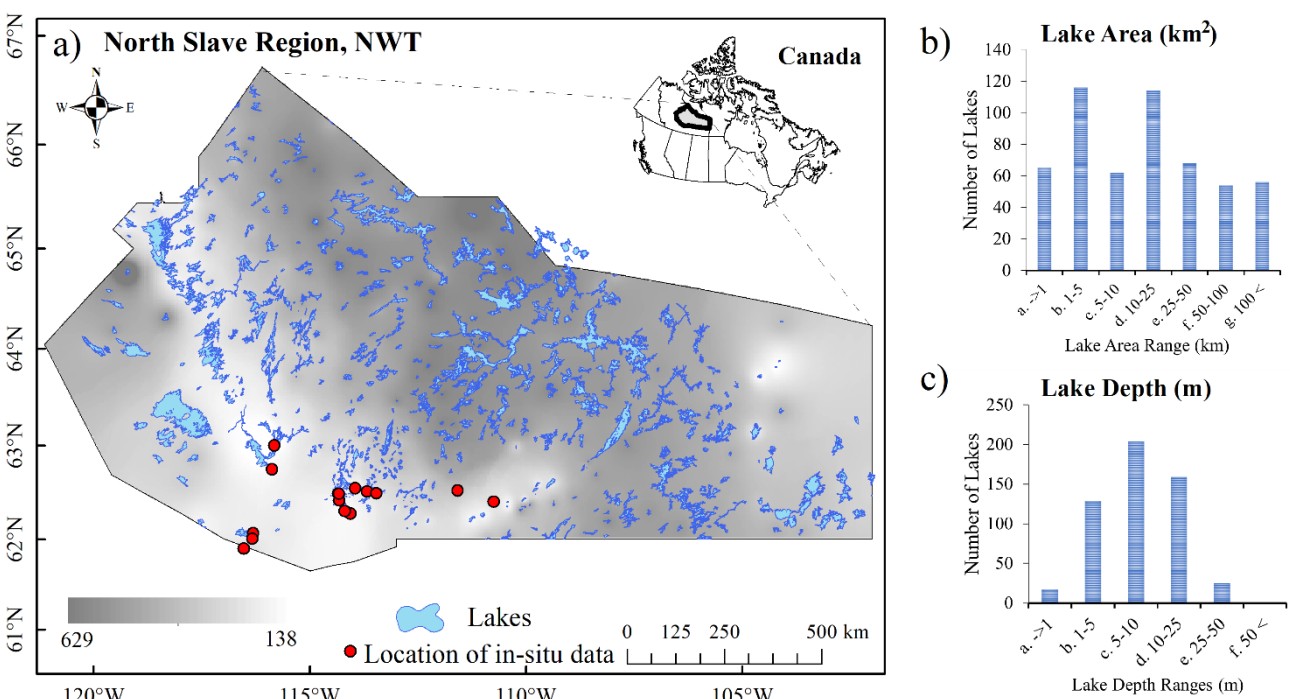

**Figure 1: Geographic distribution of study lakes in the North Slave Region, Northwest Territories, Canada. Distribution of lakes area and depth is shown in b) and c) respectively.**



## 2.2 Spatial Data for LST Retrieval

### 2.2.1 Landsat Archives

Landsat archives consists of optical data derived from a series of earth-observing satellite missions. For this study, Landsat data was obtained from the United States Geological Survey (USGS) across the NSR. Landsat thermal bands were used to

85 estimate surface temperature on lakes from the thermal infrared (TIR) bands of Landsat-5 TM (Thematic Mapper) (1984-2013), Landsat-7 ETM+ (Enhanced Thematic Mapper Plus) (1999-present) and Landsat-8 OLI/TIRS (Operational Land Imager and the Thermal Infrared Sensor) (2013-present) instruments. Landsat instruments orbit at an altitude of 705 km, are sun synchronous and have a 16-day repeat cycle. The thermal band (band 6) of Landsat-5 and Landsat-7 record emitted radiation between the wavelengths of 10.40 µm to12.50 µm while that of Landsat-8 (band 10) records in band 10 between 10.6

90 µm to 11.19 µm. Spatial resolution of thermal bands Landsat-5 TM (120 m), Landsat-7 ETM+ (60 m) and Landsat-8 OLI/TIRS (100 m) are resampled with the cubic convolution method and distributed at a spatial resolution of 30 m to match optical bands (USGS, 2022). Other bands including the quality band (BQA), near infrared band and the red bands in addition to metadata are also used in the retrieval of LST. About 34 Landsat tiles scenes covers the NSR with each tile containing $5000 \times 5000$ 30 m pixels and overpass times ranging between 18:00 to 20:00 UTC.

### 2.2.2 ERA5 Reanalysis Data

Total column water vapour from ERA5 reanalysis data (Copernicus Climate Change Service (C3S), 2017) from 1984 to 2021 was used as input in the algorithm to correct for atmospheric effects on Landsat images. Data was derived from the hourly data with a ~30 km spatial resolution from the European Centre for Medium-Range Weather Forecasts (ECMWF) (Hersbach et al., 2020). ERA5 reanalysis data is a dataset generated from a combination of in-situ observations and modelling to provide

estimates of land, atmospheric and ocean data on a global scale. Average ERA5 hourly total column water vapour on single levels was used in the LST retrieval algorithm

### 2.2.3 Lake Outline and Properties Data

For each lake, the name, location, depth, size, elevation, and outline of the lake was retrieved from a combination of HydroLAKES database, CanVec series and the Water file-Lakes and Rivers database. HydroLAKES database is a digital map

repository developed in the Global HydroLAB (http://wp.geog.mcgill.ca/hydrolab/) from a collection of several databases (*e.g.,* Global and regional databases like CanVec series and SRTM Water Body Data (Slater et al., 2006)). This database provides information on world lakes and their major properties in the form of high-resolution maps. Over 1,427,688 individual lake vector polygons greater than 10 ha is included in the repository (Messager et al., 2016). The mode of pixel-level lake elevation data obtained from the Earth-Env-DEM90 digital elevation model and the USGS provided GTOPO30 DEM is used

to calculate HydroLAKES elevation data. A geostatistical model was used to derive average depths and volumes for lakes, derived from surrounding land surface topography (Messager et al., 2016). As part of the Government of Canada initiative

(https://open.canada.ca), CanVec series provides geometric description and fundamental characteristics of hydrographic phenomena in the form of geospatial vector data. The Water file-Lakes and Rivers polygons data (https://www12.statcan.gc.ca) maps lakes and rivers under the 2006 census, created by statistics Canada under the Government of Canada on August 29,
2013. This data was the major source of lake names attributed to lake outlines in our dataset.

### 2.2.4 Evaluation Dataset

Landsat-derived LST was generated during both open water and ice-covered season. Retrieved data were evaluated against in-situ measurements collected over selected locations within the study area (Figure 1). In-situ measurements from Mackenzie DataStream was used for evaluating LST derived from Landsat. DataStream is an open access freshwater data platform that
provides water monitoring data collected by governments and communities across Canada (Environment and Climate Change Canada, 2020). The database for the NWT region was the product of NWT-wide community-based water quality monitoring (CBM) program, which are collected during open water seasons. The CBM program was implemented in 2012 as a partnership between the Department of Environment and Natural Resources (ENR), Government of the Northwest Territories (GNWT), communities and regional organizations in NWT with the aim of monitoring water quality and changes. Surface temperature
of lakes were measured with YSI Sondes and EXO 2 Sondes and interpreted by ENR. Collated surface temperature data used for evaluation from this source was from the years 2014 to 2019. Another major source was lake temperature data collected by Environment and Climate Change Canada (ECCC) from 1999 to 2003. Temperature loggers were used to measure hourly temperature on lakes for given periods during open water periods, however only temperature collected at the skin surface (depth= 0 m) was used for LST evaluation in this study.
MODIS (MYD11_L2) surface temperature dataset from 2003 to 2021 was used to evaluate Landsat-derived LST data generated during both open water and ice-covered seasons. The dataset was obtained from NASA's Earth Observing System Data and Information System (EOSDIS). Mounted on terra and aqua satellites, MODIS records within the spectral ranges of 0.405 - 14.385 μm across 36 bands. The aqua product contains nighttime and daytime LST measurements on a spatial resolution of ~1 km derived from the thermal infrared bands. For this study the daytime LST measurement covering lakes in the NSR
were compared against the Landsat-derived LST.

## 3. Methods

### 3.1 Algorithm for Lake Surface Temperature

The thermal bands of Landsat were used in the retrieval algorithm to generate North Slave LST (band 6 for Landsat-5TM/Landsat-7ETM+ and band 10 of Landsat OLI/TIRS). Atmospheric and emissivity correction of thermal bands were
conducted to account for the effect of absorption and emission on surface radiation. A single channel (SC) method was adapted and applied in this study for the retrieval of LST (Jimenez-Munoz et al., 2009, 2014; Jiménez-Muñoz & Sobrino, 2003). This method is based on approximating the radiative transfer equation without the dependence on in-situ radio sounding data. A





single band is used in the SC method making is feasible for single thermal band satellites like Landsat-5 TM which was used in this study. SC method uses atmospheric water vapour (Sect. 2.2.2) as a variable in the correction for atmospheric effect.

LST retrieval using the SC method requires atmospheric water vapour, emissivity, brightness temperature and wavelength emitted radiance values in addition to thermal constants. LST estimation is based on the following Eq. (1) (Jiménez-Munoz & Sobrino, 2003):

$$LST = \gamma\left[\varepsilon^{-1}\left(\psi_1 L_{sensor,\lambda} + \psi_2\right) + \psi_3\right] + \delta,\qquad(1)$$

where:

$$\gamma = \left\{\frac{c_2 L_{sensor,\lambda}}{T_{sensor}^2}\left[\frac{\lambda^4}{c_1} L_{sensor,\lambda} + \lambda^{-1}\right]\right\}^{-1},\qquad(2)$$

and:

$$\delta = -\gamma L_{sensor,\lambda} + T_{sensor},\qquad(3)$$

At-sensor radiance and brightness temperature are denoted by $L_{sensor,\lambda}$ (W m$^{-2}$ sr$^{-1}$ μm$^{-1}$) and $T_{sensor}$ (K) respectively. $c_1$ (1.19104 10$^8$ W μm$^4$ m$^{-2}$ sr$^{-1}$) and $c_2$ (14387.7 μm K) are Plank's constants. Emitted radiance wavelength ($\lambda$) is 11.457 μm

in Landsat-5 TM, 11.269 μm in Landsat-7 ETM+ and 10.904 μm in Landsat-8 OLI TIRS. $\psi_1$, $\psi_2$ and $\psi_3$ are atmospheric functions obtained as a function of water vapour ($w$) and are specific to the three individual Landsat sensors.

At-sensor spectral radiance were calculated from raw digital numbers (DN) of thermal bands based on metadata information and constants. Equations used are specific to the type of sensor as listed below.

At-sensor radiance values for Landsat-5 TM was derived using Eq. (4) (Chander & Markham, 2003):

$$L_{sensor,\lambda} = G_{rescale}.DN + B_{rescale},\qquad(4)$$

where 0.0551584 Wm$^2$sr$^1$μm$^1$/DN and 1.2378 Wm$^2$sr$^1$μm$^1$/DN are constants for $G_{rescale}$ and $B_{rescale}$ respectively.

Landsat-7 ETM+ was derived using Eq. (5) (Ihlen & Zanter, 2019) :

$$L_{sensor,\lambda} = \left(\frac{L_{\lambda max} - L_{\lambda min}}{Q_{calmax} - Q_{calmin}}\right)(Q_{cal} - Q_{calmin}) + L_{\lambda min},\qquad(5)$$

where the maximum and minimum spectral radiance is represented by $L_{\lambda max}$ and $L_{\lambda min}$ respectively and the maximum and minimum quantized calibrate pixel is represented by $Q_{calmax}$ and $Q_{calmin}$ is respectively, obtained from the metafile. DN values of pixels in band 6 is denoted by $Q_{cal}$.

Landsat-8 OLI TIRS was derived using Eq. (6) (U.S. Geological Survey, 2016):

$$L_{sensor,\lambda} = M_L Q_{cal} + A_L,\qquad(6)$$



where DN values of pixels in band 10 is denoted by $Q_{cal}$. $M_L = 0.000342$ and $A_L = 0.1$ are fixed rescaling factor provided by the USGS in the metadata data.

Brightness temperature $T_{sensor}$ is estimated using calculated at-sensor radiance values and thermal constants derived from the metadata based on Eq. (7) below:

$$T_{sensor} = \frac{K_2}{ln\left(\frac{K_1}{L_{sensor,\lambda}}+1\right)}, \tag{7}$$

where thermal constants $K_1$ (W m$^{-2}$ sr$^{-1}$ μm$^{-1}$) and $K_2$ (K) vary based on type of Landsat sensor (Table 1).

**Table 1: Thermal constants applied to Landsat thermal bands for brightness temperature estimation**

| Thermal Constant | Landsat-5 TM Band 6 | Landsat-7 ETM+ Band 6 | Landsat-8 OLI/TIRS Band 10 |
|---|---|---|---|
| $K_1$ | 607.76 | 666.09 | 774.8853 |
| $K_2$ | 1260.56 | 1282.71 | 1321.0789 |

Atmospheric Functions (AFs) used for atmospheric correction were based on coefficients acquired using Global Atmospheric
Profiles from Reanalysis Information (GAPRI) and Thermodynamic Initial Guess Retrieval (TIGR) databases (Jimenez-Munoz et al., 2009, 2014).

Atmospheric Functions Equations $\psi_1$, $\psi_2$ and $\psi_3$ particularized for Landsat-8 OLI/TIRS 8 are:

$$\psi_1 = 0.04019w^2 + 0.02916w + 1.01523, \tag{8a}$$

$$\psi_2 = -0.38333w^2 - 0.50294w + 0.20324, \tag{8b}$$

$$\psi_1 = 0.00918w^2 + 1.36072w - 0.27514, \tag{8c}$$

Landsat-7 ETM+ AFs:

$$\psi_1 = 0.07593w^2 - 0.07132w + 1.08565, \tag{9a}$$

$$\psi_1 = -0.61438w^2 - 0.70916w - 0.19379, \tag{9b}$$

$$\psi_1 = -0.02892w^2 + 1.46051w - 0.43199, \tag{9c}$$

Landsat-5 TM AFs:

$$\psi_1 = 0.07518w^2 - 0.00492w + 1.03189, \tag{10a}$$

$$\psi_1 = -0.59600w^2 - 1.22554w + 0.08104, \tag{10b}$$



$$\psi_1 = -0.02767w^2 + 1.43740w - 0.25844 , \tag{10c}$$

Normalized Difference Vegetation Index (NDVI) (Eq.8) values calculated were used to assign surface lake surface emissivity.
Infrared (NIR) and red bands of Landsat was used to calculate NDVI values with (Eq.11).

$$NDVI = \frac{NIR - Red}{NIR + Red} , \tag{11}$$

The lake surface was assigned an emissivity of 0.985 if NDVI values were lower than 0.05, otherwise a value of 0.97 was assigned (Prats et al., 2018).

### 3.2 Retrieval of Lake Surface Temperature

**3.2.1 LST Retrieval**

LST retrieval algorithms were applied to the thermal bands in conjunction with other processed output from Landsat data to generate the LST dataset. Quality Assurance (QA) band outlining surface, atmosphere, and sensor conditions included in the Landsat data were used to mask out clouds and other obstructions. The QA band assesses cloud influence at different confidence levels [high (67-100 %), medium (34-66 %) and low (0-33 %)] making it possible for cloud removal. In this study,
high and medium confidence values were categorized as cloud pixels while low confidence was considered cloud free pixels. LST retrieval algorithms and equations (Eq. 1 – Eq. 11) were applied to thermal bands of all tiles from 1984 to 2021. Cloud masks were generated and applied to retrieved LST to eliminate cloud distorted pixels. LST pixels were extracted using vector files of lake outline from the HydroLAKES datasets. A 100 m negative buffer was applied to remove the effect of lake pixel mixing with land surface pixels. Possible erroneous pixels were flagged using z-scores which calculate how far a value is from
the mean and were used to access spatial differences and outliers in pixels. Pixels with z-score values of above 3.5 and below -3.5 of lakes were flagged. LST output with equal pixels across the entire lake or group of pixels having the same value to four decimal places were flagged. Further visual quality checks and analysis were applied to flagged LST to clean generated the data and remove erroneous cloud cover that could not be captured in masks. The overall framework for retrieval and generation of LST dataset for selected lakes in the NSR is highlighted in Figure 2.



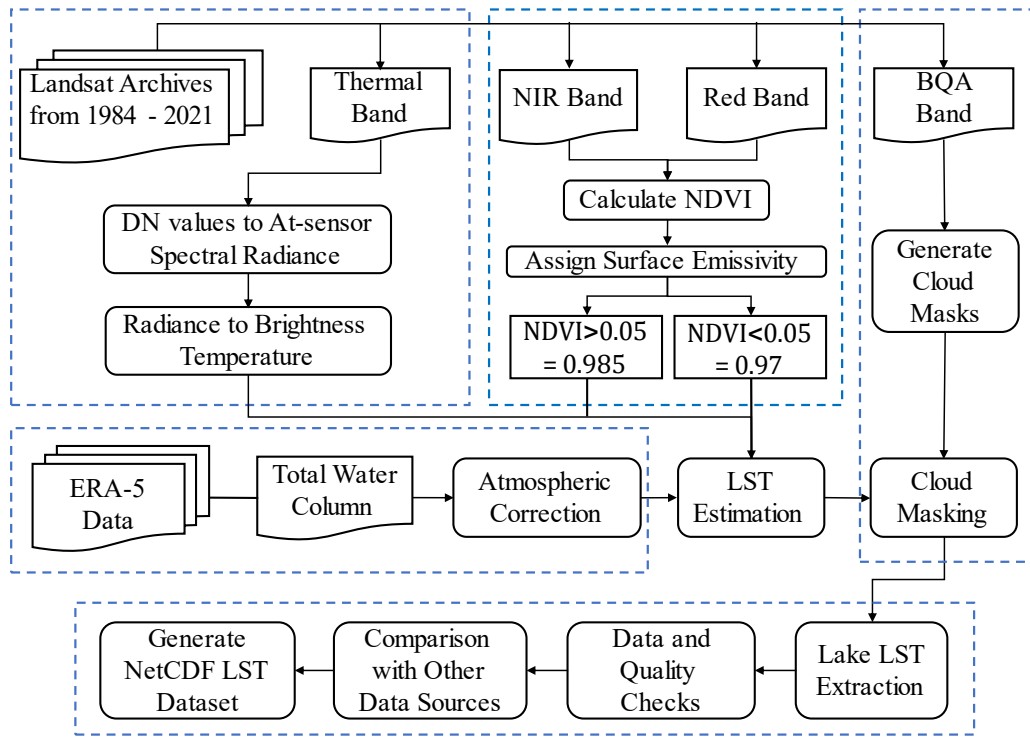

**Figure 2: Workflow and methods for generating LST dataset from Landsat archives.**

### 3.2.2 Data Quality Assessment Information

It is important to highlight the limitations in data estimates from satellite-based records (Merchant et al., 2017). This provides

awareness to the degree to which a sensor is stable as well as observations obtained from them. These reports are necessary to

inform the confidence of data extracted and the structures of their errors through time and space. One major distortion of

Landsat archives is the failing of the scan line corrector of Landsat-7 ETM+ on 31st May 2003. Measurement from scans could

not be corrected rendering all images sensed after that date losing about 22% of data extracted. This limitation named Landsat-

7 ETM+ SLC-off issue is more prominent in the edges of images than the centre. Landsat-7 ETM+ data was still used in the

study because the radiometric and geometric corrections are unaffected by this scan line issue.

### 3.2.3 Evaluation Methods

Indicators used to evaluate the performance of North Slave LST against in-situ and MODIS LST were the root mean squared

deviation (RMSD), mean bias deviation (MBD) and R-squared. The MBD, which assesses systematic differences, evaluates

the under prediction and over prediction between two datasets. An MBD value of 0 indicates a completely random error.



$$MBD = \frac{\sum_{i=1}^{N}[P_i - O_i]}{N} \, , \qquad\qquad (12)$$

where $O_i$ and $P_i$ are the observed and predicted values respectively while $N$ is number of points used for evaluation. Values of the index ranged between 0 and 1 indicating the worst and best possible performance respectively.

The root mean squared deviation (RMSD) measures total difference between two datasets without distinguishing between over or under prediction of models/algorithms. No deviation in values result in an RMSD value of 0.

$$RMSD = \sqrt{\frac{\sum_{i=1}^{N}[P_i - O_i]^2}{N}} \, , \qquad\qquad (13)$$

**4 Results and Discussion**

**4.1 Quality of Landsat-derived Lake surface temperature**

The main sources of limitation for North Slave LST products include (i) potential mixed pixels that might not captured by the algorithm (ii) presence of *no data* pixels on lakes and (iii) inconsistency in temporal resolution of dataset per lake. Lake boundaries extraction of LST was based on outlines from external boundary files (Sect. 2.2.4) and as such errors that may exist including overestimating lake area and incapability to demarcate lake islands accurately would affect LST values retrieved. A
240 100-meter inward buffer was applied to address this, however valuable lake shore LST information is lost especially in small lakes. The number of pixels and the percentage of the lake it represents is reported in Appendix A. Depending on lake shape, area and existence of islands, pixels represented 16.7% to 97.34% of lake area. The spatial variation in LST is reduced for lakes with smaller number of pixels.

In addition to the overall representativeness of pixels on lakes LST pixels retrieved for a given day may vary due to cloud cover and Landsat-7 ETM+ SLC-off issue (Sect. 3.2.2). This results in missing LST pixels for a given lake. These pixels are represented with *no data* pixels (pixels which do not contain LST values) in dataset. Figure (3) highlights the fraction of LST pixels to *no data* pixels distributed over years and months. The percentage of *no data* pixels ranged from 30.6% (1996) to 45.4% (1993) across the years with relatively lower *no data* pixels percentages recorded from 2014 to 2021 (less than 37.2%)
(Figure 3a). Generally, earlier years recorded higher *no data* pixels percentages compared to later years. Monthly distribution (Figure 3b) showed the least percentage of *no data* pixels for the month of February (26.8%) and the highest for the month of October (51.2%).





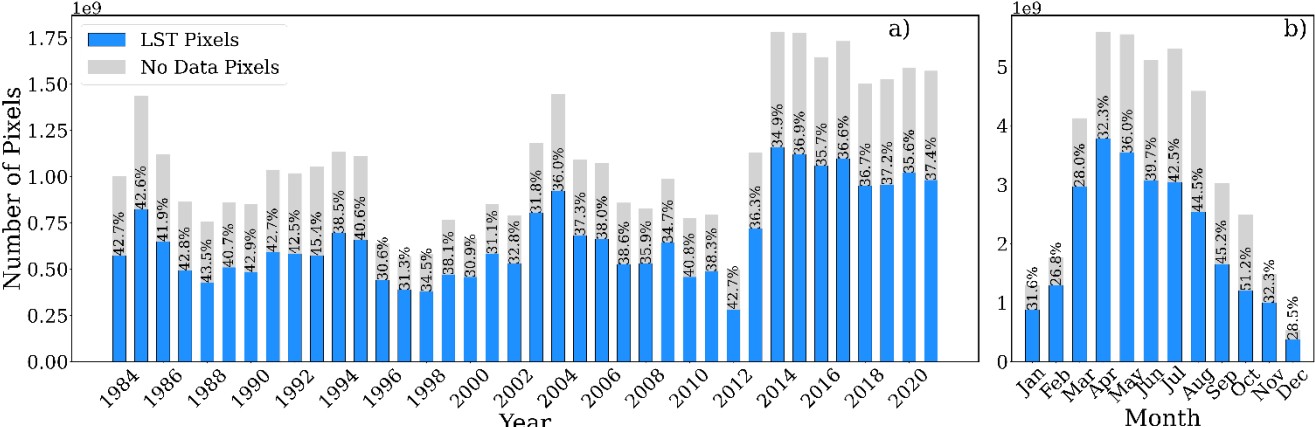

**Figure 3: a) Year and b) monthly distribution of LST pixels vs *no data* pixels. Highlighting the percentage, *no data* pixels for a given period.**

Due to the presence of *no data* pixels, it is necessary to inform on the percentage coverage of LST pixels. LST pixel coverage for each image is calculated as - LST pixels retrieved divided to the number of total pixels for a given lake multiply by 100 percent. LST pixel coverage is reported for each lake on a given day as part of the naming and metadata of our dataset. Figure (4) shows the yearly distribution of LST pixel coverage for the entire dataset. Lakes with less than 10% of LST pixels on a given day were eliminated from the dataset. The percentage of lakes with LST pixel coverage greater than 90% was 47.2% (Figure 4a). A greater percentage (77.4%) of lakes in the dataset had more than 50% LST pixels coverage. The percentage of lakes with LST pixels coverage greater than 90% is plotted in Figure 4b on an annual basis. Results show a general reduction in percentage with time, where earlier years had higher percentages of LST pixels coverage > 90% than in in recent years. This downward trend can be attributed to the Landsat-7 ETM+ SLC-off issue, which increases the presence of *no data* pixels.

Even though the typical overpass for Landsat is 16 days, temporal resolution of the North Slave LST dataset varied due to overlap of satellite sensors for certain years and the inability to retrieve LST due to the cloud cover. The distribution and frequency of the data was based on the operational times of the three Landsat satellite used in this study. Majority of the LST dataset was derived from Landsat-5 (43%). Landsat-7 and Landsat-8 contributed to 34% and 22% of the dataset respectively. LST images from 1999 were derived from two set of Landsat (Landsat-5 and Landsat-7 from 1999 to 2011) and (Landsat-7 and Landsat-8 from 2013 to 2021). Years with overlapping sensors may have shorter temporal resolution compared to year with only one sensor retrieval. As a result of this there is an inconsistency with the temporal resolution of LST product.


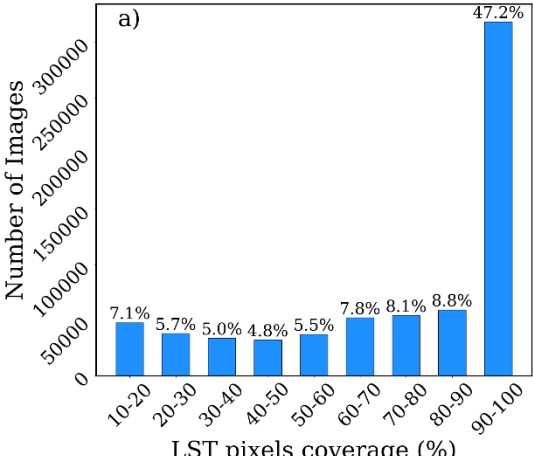

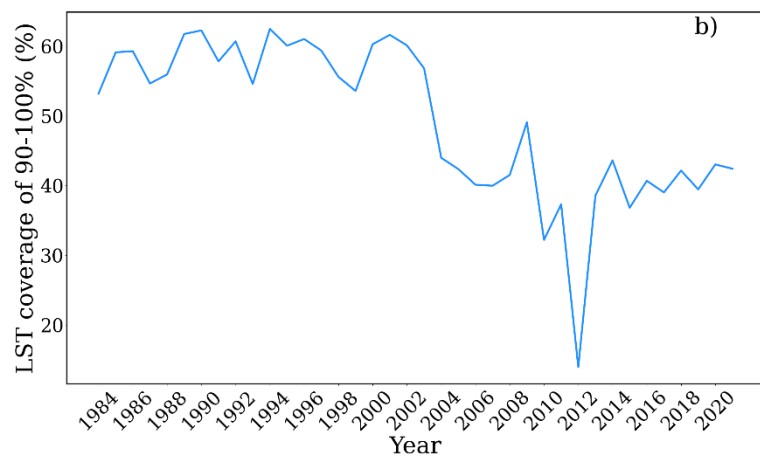

Figure 4: a) Distribution of LST pixels coverage (%) and b) yearly percentage of dataset with LST coverage ranging from 90- 100%.

## 4.2 North Slave LST dataset Evaluation

### 4.2.1 Evaluation of LST Data

The accuracy of generated North Slave LST were examined by evaluating Landsat-derived LST to corresponding in-situ data (Fig 5). Dates from both measured in-situ surface water temperature data (DataStream and ECCC) and derived North Slave LST data were matched up. In addition, comparison with the generated dataset was conducted using the widely used daily MODIS LST. Ground-based observations were compared against equivalent pixel within which measurements were taken and North Slave LST data were plotted against corresponding in-situ surface temperature measurements (Figure 5). A good correlation was observed between North Slave LST data and in-situ surface water temperature, with an $R^2$ value of 0.89 for the regression line. North Slave LST were slightly higher than in-situ records with an MBD of 0.12 and RMSD of 1.71 °C.

Deviations between North Slave LST and measured surface water temperature could be due to differences between image acquisition times and the time of the in-situ measurements. Landsat capture times of the NSR ranged between 18:00 and 20:00 UTC corresponding to 12:00 - 14:00 local time. Time of in-situ observations however were variable and did not necessarily correspond to the time of satellite image acquisition. Further variations in North Slave LST can also be attributed to the differences in sample collection as well as spatial resolution, where North Slave LST is essentially the mean of ~60 to 120 m² area as opposed to a single in-situ location. Possible errors reported by other studies for the differences in measured and Landsat values includes georeferencing, radiometric and memory effects (Chander & Markham, 2003; Markham et al., 2014; Sentlinger et al., 2008; USGS,2022, Young et al., 2017).


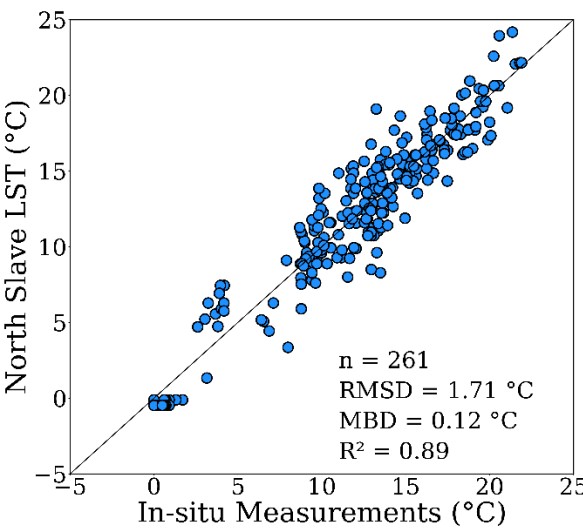

**Figure 5: Comparison of North Slave LST with DataStream and ECCC in-situ measurements of water surface temperature during open water seasons.**

**4.2.2 Yearly and Monthly Comparison of LST data to MODIS Data**

MODIS LST was first compared against available water surface temperature measurements from DataStream (Figure 6a) and Landsat-derived LST for days when records were available from all three data sources. The aim was to compare the deviation of Landsat-derived LST and water surface temperature to that of MODIS and water surface temperature. A relatively low coefficient of determination was observed for MODIS LST ($R^2$= 0.5) compared to Landsat-derived LST ($R^2$= 0.94) when evaluated against water surface temperature. RMSD values were also higher for MODIS LST (4.63 °C) than North Salve LST data (1.55°C) with MBD of 2.35°C and -0.12°C for MODIS and North Salve LST, respectively.

LST dataset was further compared against MODIS from 2003 to 2021 (ice covered and open water separately) for larger study lakes (30 km$^2$) to avoid pixel mixing with land (Figure 6b). Results showed an RMSD of 2.56°C and MBD of 1.45°C for ice covered LST suggesting an over estimation of Landsat-derived LST during this period. An under estimation was observed (MBD = -1.14°C) for open water LST with a relatively higher RMSD of 3.39. This was expected as overestimates LST when compared against in-situ data (Figure 6a). Even though prior comparison of MODIS LST to surface water temperature demonstrated a lower coefficient of determination, Landsat-derived LST was still further compared against MODIS LST in this study. The decision to use MODIS for comparative analysis however was due to unavailability of continuous historical measurements of lake surface temperature. Additionally, MODIS LST provided an added outlook on the capability of Landsat-derived LST to highlight historical trends despite low temporal resolution by demonstrating a good correlation between them LST values (R = 0.93).





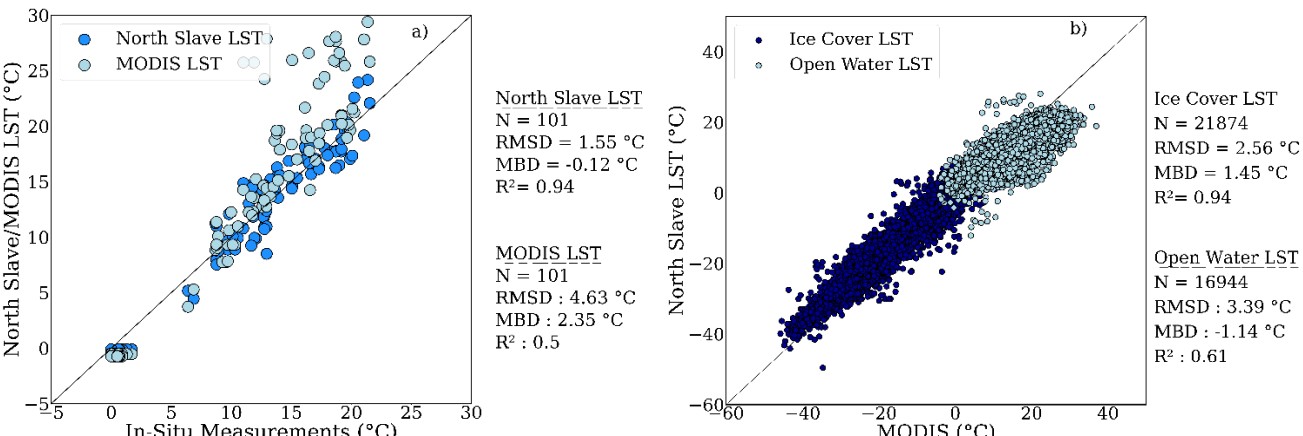

**Figure 6: Comparison of North Slave LST and MODIS LST to a) DataStream and ECCC in-situ water surface temperature measurements during open water seasons b) MODIS LST during open water and ice-covered seasons.**

Figure 7a and 7b demonstrates the yearly and monthly RMSD values derived from comparison between North Slave LST and MODIS LST. Yearly RMSD show a generally decreasing RMSD from earlier years to the later years. This may be attributed to the Landsat's sensor change in the recent years. LST values derived from 2013 onwards were extracted from Landsat-8 OLI/TIRS, which is known to have improved signal to noise ratio and calibration, higher 12-bit radiometric resolution and narrower spectral bands compared to previous sensors (Irons et al., 2012; Roy et al., 2014). Most importantly Landsat-8 OLI is known to have a radiometric uncertainty of 3% compared to that of Landsat-7 ETM+ (5%), as well as reduced band saturation (Markham et al., 2014). Monthly RMSD comparing MODIS data to generated LST decreased showed RMSDs were lowest in spring and highest in winter. LST in spring months (March – May) had the lowest RMSD (1.9°C - 2.9°C the least deviation compared with MODIS data.

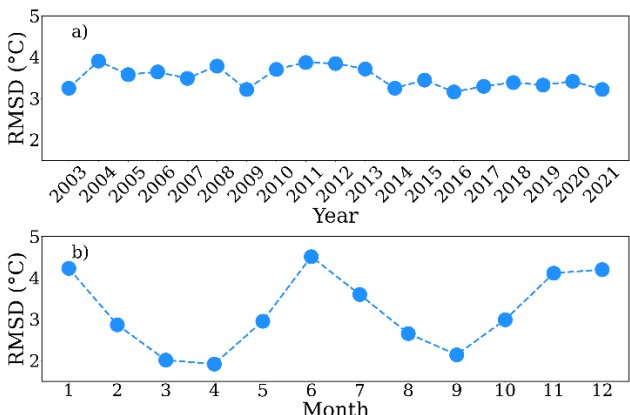

**Figure 7: Yearly and monthly RMSD values and mean bias from evaluating North Slave LST against MODIS LST from 2003 to 2021.**



### 4.3 LST Dataset Distribution

**4.3.1 Temporal Dataset Distribution of LST dataset**

LST dataset is derived from the thermal radiation of the uppermost layer of lakes hence the skin temperature. A total of 673,223 gridded data files were included in the generated North Slave LST dataset for the 535 lakes studied across the NSR. The yearly and monthly distribution of the dataset within and between lakes varied temporally, which is highlighted in Figure 8. Overall, yearly distribution of North Slave LST dataset was greater in recent years with the period between 2014 to 2021 having majority of the data and percentages ranging from 4.15 – 5% of the total dataset. Larger data files in recent years were due to LST retrieval from a combination of Landsat-7 and Landsat-8 compared to a single sensor retrieval (Landsat-5) for earlier years. Highest yearly percentage of the North Slave LST dataset was for the year 2014 (5%) and the least was for 1988 (1.2%). The bulk of unavailable data for the various years was predominantly because of insufficient usable Landsat data for winter months.

Monthly distribution of North Slave LST dataset showed the month of May with the highest percentage (13.9%) and December (1.3%) with the lowest. Generally colder months (October – April) had less data (42.7%) compared to relatively warmer months (May – September) (57.3%). Data is unevenly distributed across months and years due to differences in overpass times and influences like cloud cover and other atmospheric impact on data retrieval.





**Figure 8: Yearly and monthly distribution of North Slave LST dataset from 1984 to 2021. Percentages (%) represent the total percent of the entire data for each month or year.**


### 4.3.3 Spatial Dataset Distribution of LST dataset between lakes

While the lakes are widely distributed across the NSR, a large number (144 out of 535) captured in our dataset were within
150 km distance of Yellowknife. Yearly average number of images for each individual lake in the study region is demonstrated
in Figure 9. Average yearly minimum number of images for each lake was 20 and reached a maximum of 45. Lakes with
relatively smaller number of images were mainly distributed around Yellowknife. Smaller size lakes generally had a smaller
number of images compared to relatively larger sized lakes. This can be attributed mainly to cloud cover covering the entirety
of small lakes. Majority of lakes (152 out of 535) had between 40 and 45 images and 71% of the total lakes in the dataset had
more than 30 images per year. Lakes with lower number of pixels have a higher likelihood of being entirely cloud covered and
lose relatively more surface area due to the lake buffer.

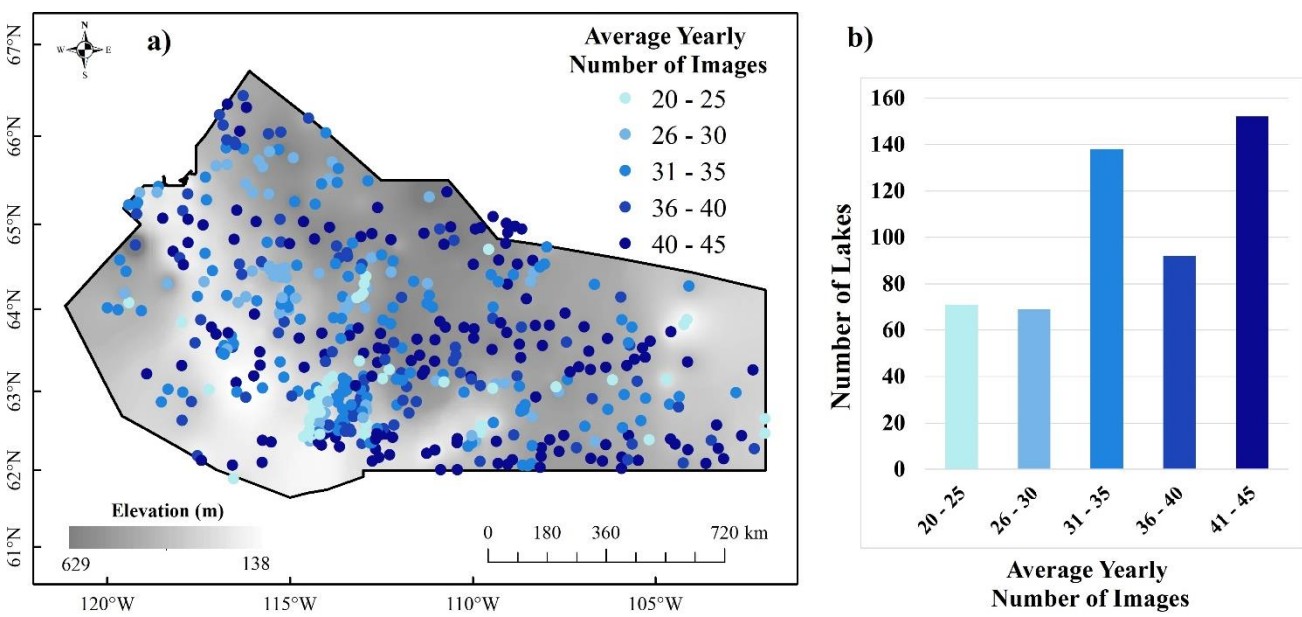

**Figure 9: Distribution of average yearly number of images (available images, or useable images) for lakes in the NSR. Can you incorporate lake size somewhere in the right panel here, or would that be too messy?**

### 4.4 North Salve LST dataset

North Slave LST dataset generated includes LST for 500 lakes with known names and 35 without. The datasets are provided
as individual NetCDF (Network Common Data Form) files which is a file format for storing multidimensional data. Spatial
coverage and dimension of LST for the study lakes are captured in this dataset. To facilitate easy query of the data each
NetCDF filename includes the name of lake, date, longitude, latitude, minimum, maximum and mean LST, number of pixels
and the percentage area of the lake LST pixels cover for a given day. Naming convention for lakes and their explanation is
summarized in Table 2. The dataset was grouped based on the name of the lake and further into yearly sub-groups.



The NetCDF files in our dataset has a two-dimensional variable "lst" which shows the spatial distribution of lake surface temperature. In addition is the one-dimensional x and y that shows the extent of the lake and number of pixels. Spatial reference for the data is the World Geodetic System 1984, EPSG:4326.

**Table 2: Sections of LST Dataset NetCDF filename and Explanation.**

| Sample File name: AcastaLake_19840428_ -115.564 _ 65.3783_-5.90_ -7.10_-6.50_17482_099.nc | |
|---|---|
| **Section of Name** | **Explanation of the section** |
| **Lake name:** AcastaLake | Name of lakes were predominantly derived from the Water file-Lakes and Rivers polygons data from Statistics Canada. Lakes unknown names were prefixed "NoNameLake" and a number. |
| **Date:** 19840428 | The date in the NetCDF file is in the format "YYYYMMDD" and represent the corresponding date the Landsat scene was captured. |
| **Longitude** (°): -115.564 | The longitude represents a known longitude predominantly located at the centre of a lake when plotted against the latitude in decimal degrees. |
| **Latitude** (°): 65.3783 | The latitude represents a known latitude predominantly located at the centre of a lake when plotted against the longitude in decimal degrees. |
| **Maximum Temperature (°C):** -5.90 | This is the maximum LST value retrieved from a lake for a given date which is the coldest part of the lake |
| **Minimum Temperature (°C):** -7.10 | This is the minimum LST value retrieved from a lake for a given date which is the warmest part of the lake |
| **Mean Temperature (°C):** -6.50 | Mean LST value calculated from the number of LST pixels retrieved for a lake on a given date. |
| **Number of LST Pixels:** 17482 | Number of LST pixels retrieved on a given lake for a given date. |
| **LST pixels coverage (%):** **099** | Number of LST pixels retrieved the lake for a given date divided by the total number of pixels representing the lake. |

## 4.5 Spatial Patterns of North Slave LST

### 4.5.1 Seasonal Lake Spatial Distribution of North Slave LST

The spatial seasonal distribution of mean LST from 1984 to 2021 is shown in Figure 10 with the aim of highlighting the spatial variation of LST for different seasons. The distribution of average LST was computed for winter (December - January), spring (March-May), summer (June – August) and autumn (September – November) for all study lakes. LST on lakes in the NSR are generally negative in winter (-26 – -18°C) and spring (-17 – -3). Lakes are ice covered during these two seasons constituting to negative LST values. Autumn was characterized by both positive and negative LST values (-8 – 3°C). Lakes start to freeze

in autumn and the rate of freezing is influenced by several factors resulting in differences in the open water duration which affect average temperature. Average LST for summer values ranged from 6 - 22°C. Average LST ranges between lakes were

the lowest in the winter (Figure 10a) with a variability if 8 °C. The largest LST variability between lakes however for summer was twice that of winter (16°C). This is expected as temperatures on lakes during this season are influenced by several factors including lake size, elevation, depth, latitude, longitude and volume (O'Reilly et al., 2015; Xie et al., 2022) in addition to air

temperature. Seasonal LST spatial distribution provides an insight into the climate patterns of the NSR region.

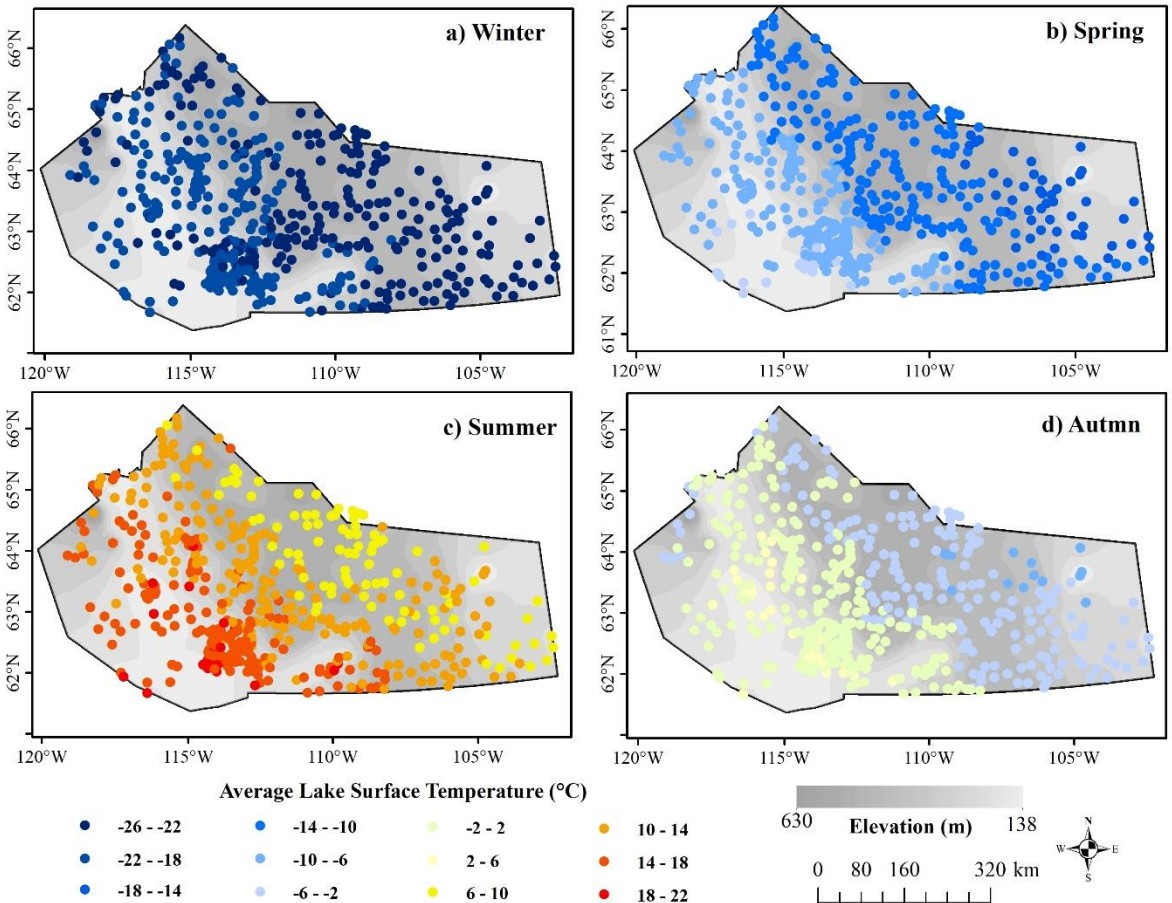

**Figure 10: Spatial distribution of average LST across the NSR showing taken across all years for a) winter, b) spring, c) summer and d) autumn.**

**4.5.2 Lake Spatial Distribution of LST for 2021**

The spatial distribution of the mean annual LST across the NSR for 2021 is shown in Figure 11a which highlights remarkable spatial differences between lakes at higher versus lower elevations, with lower elevation lakes generally demonstrating higher LST. Based on the mean annual LST values in 2021, the LST category was divided into five different ranges, as shown in the map [-12 -9°C, -9 - -6°C, -6 - -3°C, -3 - 0°C, and 0 - 3°C]. Figure 11 b shows most lakes (28%) with a mean of -3 - 0. Lake distribution in relation to mean temperature was 8%, 22%, 27%, 28% and 15% from colder to warmer LST categories,

respectively. Percentage of total area covered by lakes in relation to mean LST was 34%, 27%, 18%, 19% and 2% respectively





(Figure 11b). Although the number of lakes with LST ranging from -12 -9°C was the least (8% of lakes), the percentage of total area covered by lakes with this LST range was the largest (34% of lakes). Total area covered for all lakes with mean LST from 0 - 3°C was only 2%. This suggests that several of the lakes with warmer temperatures were smaller in size. Generally, relatively warmer lakes were also distributed around Yellowknife and the southwestern part of the region.

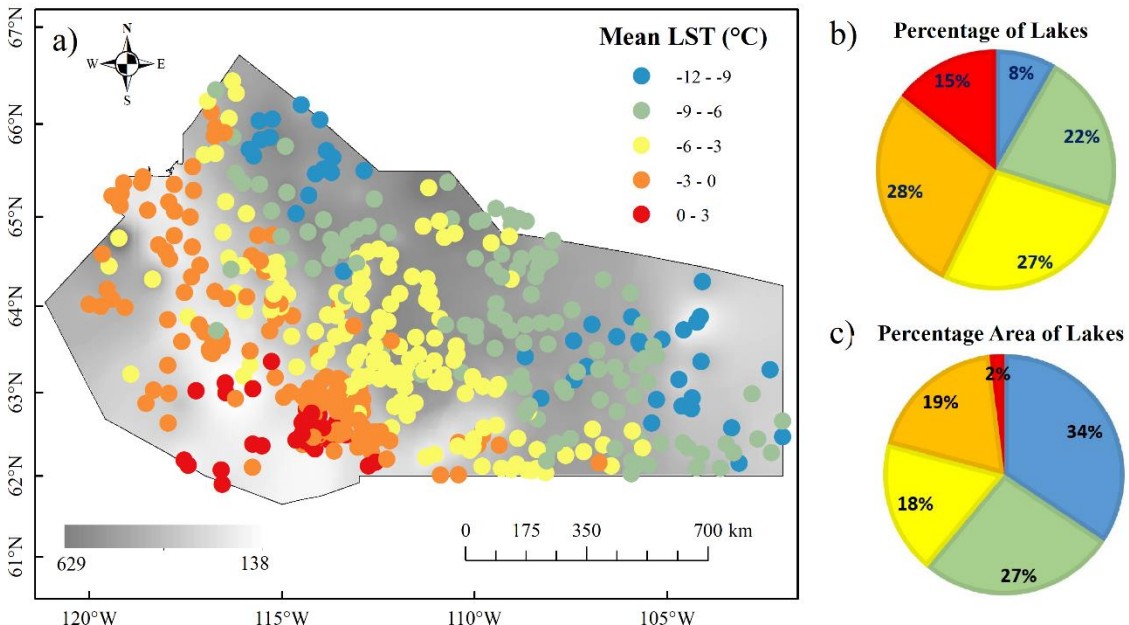

**Figure 11: Spatial distribution of mean LST for the year 2021 across the NSR showing b) the percentage number of lakes and c) percentage area of lakes within specific LST ranges.**

### 4.5.3 Intra Lake Spatial Distribution of generated LST

Lakes in several studies are treated as a homogenous entity, however, for a given lake there is spatial variability in the surface temperature based on several factors including difference in morphometry or biological, physical, and anthropogenic activities occurring on the lake at a given time (Crosman & Horel, 2009; Huang et al., 2017; Selman & Misra, 2014; Yang et al., 2020). In view of this, the North Slave LST datasets generated in this study can highlight the spatial variability within a given lake. As expected, the high spatial resolution and multidate LST generated show the heterogeneity of surface temperature of lakes. The phenomena have been demonstrated with LST on the 9th of July 2021 for a few selected lakes within our study as examples (Figure 12).

Lakes may demonstrate significant surface temperature variations for various reasons including wind redistribution, depth, biological and anthropogenic activities. In general, warmer LST are generally at the shallower coastal regions of lakes however internal LST variations differ. An example is in the case of Lake Duncan (Figure12), which demonstrated warmer temperature at the north part of the lake than the south. Maximum and minimum LST on lakes also differ with some lakes having wider



variations (*e.g.*, Duncan Lake (24 – 14°C) and Frame Lake (31 – 24°C)). Lakes physical differences as well as the location and elevation may contribute to the different ranges of surface temperature distribution on individual lakes.

**Figure 12: Intra lake spatial distribution of LST on selected lakes in the NSR highlighting the ability of the dataset to capture small scale details of LST.**

**5 Data availability**

The long-term (1984 -2021) continuous high resolution (30 m spatial resolution) regional (North Slave region, NWT) gridded LST dataset is available at https://doi.org/10.5683/SP3/J4GMC2 (Attiah et al., 2022) and the Government of Northwest Territories' (NWT) Discovery Portal (DOI will be made available at the end of the publication process). Additional data used in this study include the Landsat imagery can be downloaded from the USGS platform. Physical properties and names of lakes

were derived from HydroLAKES (https://www.hydrosheds.org/products/hydrolakes),Water file-Lakes and Rivers polygons

data (https://www12.statcan.gc.ca/census-recensement/2011/geo/bound-limit/files-fichiers/ghy_000c06a_e.zip) and CanVec series (https://open.canada.ca/data/en/dataset/9d96e8c9-22fe-4ad2-b5e8-94a6991b744b). Evaluation data was derived from Mackenzie DataStream (https://mackenziedatastream.ca/). ERA5 reanalysis data was obtained from Copernicus Climate Change Service (https://cds.climate.copernicus.eu/#!/search?text=ERA5&type=dataset).

**6 Conclusions**

A new gridded dataset (North Slave LST) of lake surface temperature across the NSR, NWT was presented in this study based on an LST retrieval algorithm adapted to the thermal bands of Landsat archives. LST data is available for 38 years (from 1984 to 2021) on a 30 m spatial resolution with varying temporal resolution (minimum of 1 day). North Slave LST dataset has proven comparable with LST products like MODIS (1 km resolution) and other water surface temperature measurements and
435 suitable for small lakes by capturing small scale details of LST.

North Slave LST dataset generated includes 673,223 NetCDF gridded data files in total for all lakes with a greater percentage (57.3%) highlighting LST in warmer months. A high percentage (43%) of the dataset was derived from Landsat-5. Lakes had a 100-meter buffer applied to resulting in a pixel representing 16.7% to 97.34% of lake area. Majority of the dataset (77.4%) had LST pixels coverage greater than 50% out of which 42.2% had pixels coverage greater than 90%. Average yearly number
of LST files for each lake was between 20 to 45.

The retrieval algorithm applied proved successful in retrieving LST from Landsat images across the NSR with an RMSD of 1.7°C and MBD of 0.12. Dataset produced provide continuous data and highlights spatial and temporal LST of lakes in the NSR. Based on generated North Slave LST, warmer lakes are predominantly located around the town of Yellowknife and on the southwestern part of the NSR. Seasonal average LST is also highlighted using generated LST with summer having the
445 highest variation of LST (16°C) between lakes. Intra-lake variability is also highted with this dataset. The North Slave LST dataset will be continually updated with improved retrieval algorithm and up to date data as they become available.

**Acknowledgement**

This research is supported by Natural Sciences and Engineering Research Council of Canada (NSERC) Canada Excellent Research Chair- Global Water Futures (CERC-GWF) fund, Remotely Sensed Monitoring of Northern Lake Ice Using
RADARSAT Constellation Mission and Cloud Computing Processing project, Government of Northwest Territories, Environment and Natural Resources, Cumulative Impact Monitoring Program (CIMP-212), NSERC Canada Research Chair and Discovery Grant to H. Kheyrollah Pour, NSTP (Northern Student Training Program), and Cold Regions Research Centre (CRRC) at Wilfrid Laurier University.



**Author Contributions**

Gifty Attiah – Methodology, Analysis, Writing and Visualization – original draft.

Homa Kheyrollah Pour – Supervision, Resources, Writing – review & editing

Andrea Scott - Supervision, Resources, Writing – review & editing

**Competing Interests**

The authors declare no competing interest.

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



# Appendices

## Appendix A

| Lake name | Latitude (°) | Longitude (°) | Area(km²) | Elevation (m) | Average Depth (m) | Number of Pixels | Percentage of lake represented |
|---|---|---|---|---|---|---|---|
| Acasta Lake | -115.564 | 65.3783 | 18.23 | 399 | 4.7 | 17645 | 87.11 |
| Achilles Lake | -110.906 | 64.963 | 27.84 | 403 | 16.7 | 27536 | 89.01 |
| Acres Lake | -108.688 | 62.7499 | 3.36 | 333 | 9.6 | 2438 | 65.18 |
| Agassiz Lake | -112.788 | 63.1797 | 19.89 | 338 | 17.9 | 18113 | 81.95 |
| Ajax Lake | -110.58 | 64.9737 | 24.32 | 446 | 8.2 | 23524 | 87.05 |
| Alexander Lake | -108.117 | 62.2884 | 6.24 | 385 | 6.8 | 5427 | 78.21 |
| Alexie Lake | -114.083 | 62.6779 | 4.24 | 218 | 6 | 3357 | 71.23 |
| Allan Lake | -113.063 | 62.9208 | 4.35 | 273 | 8 | 3893 | 80.46 |
| Ambush Lake | -113.824 | 65.7125 | 16.02 | 413 | 13.6 | 15379 | 86.39 |
| Angelique Lake | -113.421 | 64.6265 | 17.84 | 403 | 10.2 | 16742 | 84.47 |
| Angle Lake | -114.177 | 62.8313 | 4.11 | 195 | 21.1 | 3404 | 74.45 |
| Anton Lake | -114.461 | 62.9713 | 3.34 | 253 | 8.8 | 2404 | 64.67 |
| Ardent Lake | -115.736 | 65.6577 | 14.28 | 412 | 6.2 | 14331 | 90.34 |
| Armi Lake | -114.124 | 63.7112 | 26.59 | 354 | 9.6 | 25165 | 85.18 |
| Arno Lake | -113.533 | 63.0506 | 0.11 | 0 | 0 | 43 | 36.36 |
| Artillery Lake | -107.871 | 63.1744 | 521.89 | 352 | 24.3 | 552430 | 95.27 |
| Athenia Lake | -111.516 | 63.6452 | 42.29 | 416 | 7.2 | 38069 | 81.01 |
| Augustus Lake | -116.686 | 66.3619 | 9.78 | 340 | 22.3 | 9027 | 83.03 |
| Aurora Lake | -112.921 | 64.3918 | 15.25 | 377 | 5.3 | 14892 | 87.8 |
| Awry Lake | -114.922 | 62.9506 | 26.89 | 201 | 19.7 | 25545 | 85.5 |
| Axecut Lake | -104.138 | 63.8762 | 1.95 | 165 | 5.6 | 1789 | 82.56 |
| Aylmer Lake | -108.53 | 64.1244 | 680.73 | 355 | 19.7 | 715690 | 94.62 |
| Back Lake | -109.329 | 63.8188 | 86.59 | 384 | 12.9 | 87319 | 90.76 |
| Back River | -108.275 | 64.609 | 62.66 | 333 | 15.6 | 61472 | 88.29 |
| Baldhead Lake | -113.634 | 64.6092 | 20.02 | 409 | 11.2 | 19439 | 87.41 |
| Banting Lake | -114.285 | 62.6292 | 3.86 | 171 | 10.9 | 2959 | 68.91 |
| Barnston Lake | -110.033 | 63.1483 | 12.64 | 384 | 15 | 11950 | 85.13 |
| Bartlett Lake | -118.336 | 63.0863 | 183.8 | 260 | 4.4 | 191259 | 93.65 |
| Basile Lake | -111.261 | 62.2174 | 15.19 | 171 | 20.2 | 15196 | 90.06 |
| Basler Lake | -115.945 | 63.9303 | 99.21 | 230 | 40.3 | 99610 | 90.36 |
| Baton Lake | -115.096 | 64.3761 | 1.83 | 327 | 11.8 | 1089 | 53.55 |
| Bear Lake | -114.184 | 62.3801 | 1.7 | 173 | 3.9 | 1158 | 61.18 |
| Beauparlant Lake | -112.177 | 64.5722 | 13.62 | 445 | 6.6 | 12519 | 82.75 |
| Beauregard Lake | -114.336 | 62.7216 | 1.46 | 207 | 7.6 | 1220 | 75.34 |
| Beaverhill Lake | -104.373 | 62.8032 | 121.51 | 278 | 12.9 | 129356 | 95.81 |
| Beaverlodge Lake | -118.194 | 64.6873 | 65.31 | 175 | 6.7 | 64188 | 88.46 |
| Beck Lake | -104.613 | 62.8365 | 4.82 | 282 | 1.8 | 4839 | 90.46 |
| Bedford Lake | -109.496 | 62.9993 | 25.91 | 306 | 15.8 | 23905 | 82.86 |
| Bell Lake | -114.334 | 62.8427 | 3.82 | 226 | 7.1 | 3104 | 73.04 |
| Benoit Lake | -116.251 | 66.3525 | 25.65 | 379 | 6.6 | 24857 | 87.21 |
| Bessonette Lake | -114.741 | 63.6612 | 8.57 | 296 | 10.2 | 7970 | 83.66 |
| Betty Ray Lake | -116.574 | 63.5419 | 6.07 | 196 | 6.1 | 4935 | 73.15 |
| Bewick Lake | -105.718 | 62.4994 | 85.96 | 341 | 8.4 | 83936 | 87.88 |
| Big Lake | -112.986 | 64.857 | 65.63 | 407 | 7.8 | 66426 | 91.09 |
| Big Rocky Lake | -102.294 | 62.2768 | 78.01 | 254 | 8.1 | 75539 | 87.16 |
| Bighill Lake | -114.036 | 62.5076 | 4.58 | 189 | 6.2 | 4342 | 85.37 |
| Biologist Lake | -104.087 | 64.2761 | 3.48 | 300 | 2.5 | 3336 | 86.21 |
| Birch Lake | -116.565 | 62.067 | 85.93 | 187 | 5.7 | 89784 | 94.04 |
| Bishop Lake | -116.157 | 65.5005 | 46.4 | 347 | 14.8 | 47643 | 92.41 |
| Black Lichen Lake | -116.263 | 64.4217 | 58.82 | 287 | 23.7 | 58221 | 89.07 |
| Blaisdell Lake | -113.579 | 62.7784 | 5.95 | 249 | 6.6 | 5191 | 78.49 |
| Blake Lake | -106.448 | 62.1172 | 12.76 | 389 | 5.3 | 11905 | 83.93 |
| Bodie Lake | -105.853 | 62.9572 | 14.88 | 341 | 5.6 | 14557 | 88.04 |
| Boland Lake | -115.69 | 64.5396 | 28.93 | 255 | 33.6 | 26869 | 83.44 |
| Boulder Lake | -113.074 | 63.7656 | 16.91 | 361 | 13.2 | 16609 | 88.35 |
| Box Lake | -109.423 | 63.9199 | 39.37 | 384 | 12.9 | 38395 | 87.78 |





| | | | | | | |
|---|---|---|---|---|---|---|
| Bras dOr Lake | -115.743 | 62.3927 | 33.03 | 0 | 0 | 34349 | 93.58 |
| Breadner Lake | -116.749 | 65.8623 | 27.34 | 295 | 19.7 | 27433 | 90.31 |
| Breithaupt Lake | -105.407 | 62.6386 | 19.67 | 335 | 2.9 | 18063 | 82.66 |
| Bridge Lake | -112.276 | 63.268 | 3.58 | 395 | 6.9 | 3013 | 75.7 |
| Brock Lake | -112.833 | 62.4155 | 0.17 | 255 | 3.1 | 81 | 41.18 |
| Broken Dish Lake | -116.267 | 65.8579 | 3.91 | 368 | 11.9 | 3721 | 85.68 |
| Brown Water Lake | -115.863 | 64.5998 | 51.97 | 275 | 35 | 45575 | 78.7 |
| Buckham Lake | -112.647 | 62.2974 | 30.98 | 189 | 19.6 | 28839 | 83.8 |
| Bunting Lake | -109.783 | 62.4827 | 0.23 | 235 | 2.9 | 167 | 65.22 |
| Burbanks Lake | -108.6 | 62.7652 | 0.08 | 0 | 0 | 37 | 37.5 |
| Burke Lake | -116.712 | 63.5178 | 2.37 | 170 | 7.9 | 2074 | 78.9 |
| Bustard Lake | -108.415 | 64.3342 | 7.41 | 372 | 10.8 | 7067 | 85.83 |
| Calder Lake | -115.234 | 65.8658 | 15.31 | 454 | 6.8 | 14620 | 85.83 |
| Calypso Lake | -115.844 | 65.7256 | 12.98 | 380 | 12.4 | 11701 | 81.12 |
| Campbell Lake | -106.894 | 63.2391 | 110.9 | 373 | 7.8 | 105646 | 85.73 |
| Camsell Lake | -111.185 | 63.6228 | 158.16 | 411 | 12.6 | 153622 | 87.42 |
| Carey | -102.909 | 62.2067 | 255.34 | 265 | 12.1 | 259064 | 91.24 |
| Caribou Lake | -114.023 | 62.986 | 2.78 | 245 | 4.9 | 2077 | 67.27 |
| Carter Lake | -104.303 | 62.9554 | 29.59 | 274 | 8.1 | 27590 | 83.91 |
| Cassino Lake | -119.398 | 64.0755 | 22.44 | 325 | 4.4 | 23561 | 94.47 |
| Castor Lake | -115.978 | 64.4679 | 35.74 | 284 | 28.6 | 34101 | 85.87 |
| Chan 1 Lake | -114.355 | 62.6408 | 0.41 | 236 | 2.8 | 238 | 51.22 |
| Chan Lake | -116.542 | 61.8909 | 0.84 | 239 | 6.5 | 701 | 75 |
| Chartrand Lake | -115.532 | 64.4607 | 20.79 | 336 | 16.7 | 19626 | 84.94 |
| Chedabucto Lake | -115.553 | 62.3691 | 43.01 | 193 | 5.7 | 44908 | 93.98 |
| Chelay Lake | -119.403 | 65.2223 | 1.72 | 199 | 5.6 | 1393 | 72.67 |
| Chipp Lake | -112.626 | 62.4685 | 2.9 | 270 | 2.1 | 2056 | 63.79 |
| Chitty Lake | -114.123 | 62.7149 | 2.38 | 221 | 6.2 | 1822 | 68.91 |
| Clinton Colden Lake | -107.474 | 63.9586 | 599.71 | 352 | 13.4 | 608092 | 91.23 |
| Clive Lake | -118.906 | 63.212 | 64.84 | 255 | 3.6 | 66713 | 92.47 |
| Coldblow Lake | -104.107 | 63.361 | 12.1 | 320 | 3.8 | 11726 | 87.19 |
| Cole Lake | -116.594 | 63.6731 | 9.24 | 194 | 11.9 | 8010 | 77.92 |
| Compton Lake | -109.79 | 62.5331 | 8.91 | 246 | 26.2 | 9010 | 91.02 |
| Consolation Lake | -112.797 | 62.5081 | 20.01 | 238 | 14.8 | 15423 | 69.37 |
| Contwoyto Lake | -110.506 | 65.3085 | 163.62 | 435 | 22.2 | 166125 | 91.38 |
| Cook Lake | -108.849 | 63.1595 | 49.99 | 352 | 10.2 | 45458 | 81.84 |
| Cooley Lake | -109.052 | 62.0574 | 9.33 | 336 | 10 | 8220 | 79.31 |
| Cosmos Lake | -104.224 | 63.8148 | 2.14 | 150 | 8.9 | 2052 | 85.98 |
| Cotterill Lake | -114.847 | 64.1539 | 17.93 | 334 | 10.9 | 16688 | 83.77 |
| Courageous Lake | -111.188 | 64.1657 | 228.32 | 395 | 12.6 | 232082 | 91.48 |
| Courier Lake | -111.946 | 63.5337 | 1.46 | 439 | 3.8 | 1092 | 67.12 |
| Cowan Lake | -115.274 | 63.3612 | 4.44 | 218 | 9.7 | 3373 | 68.47 |
| Crapaud Lake | -114.021 | 62.9358 | 5.87 | 225 | 5.3 | 4945 | 75.81 |
| Credit Lake | -112.492 | 64.6574 | 17.7 | 429 | 3.2 | 16740 | 85.14 |
| Creek Lake | -114.01 | 62.4733 | 0.88 | 0 | 0 | 774 | 79.55 |
| Criss Lake | -113.514 | 63.0824 | 0.11 | 320 | 2.4 | 51 | 45.45 |
| Croft Lake | -104.216 | 62.1037 | 15.74 | 337 | 6.4 | 14405 | 82.34 |
| Crooked Foot Lake | -113.554 | 64.1502 | 9.02 | 374 | 8.2 | 8807 | 87.92 |
| Cruikshank Lake | -105.357 | 63.5315 | 10.7 | 313 | 4.7 | 10138 | 85.23 |
| Danes Lake | -111.706 | 63.2228 | 2.79 | 426 | 6.5 | 2398 | 77.42 |
| DAoust Lake | -108.915 | 62.1353 | 11.25 | 333 | 10.9 | 11028 | 88 |
| Daran Lake | -115.06 | 64.0299 | 20.75 | 257 | 37.7 | 19570 | 84.87 |
| Darrell Lake | -105.65 | 63.7836 | 21.19 | 341 | 6.7 | 17950 | 76.26 |
| Dauphinee Lake | -114.721 | 63.8824 | 9.95 | 288 | 13 | 9300 | 84.12 |
| David lake | -114.378 | 62.5436 | 0.13 | 198 | 2.7 | 55 | 38.46 |
| Davis Lake | -115.439 | 64.3984 | 1.33 | 335 | 10.8 | 1029 | 69.92 |
| Day Lake | -113.504 | 62.6637 | 0.83 | 264 | 3.6 | 610 | 66.27 |
| Defeat Lake | -113.643 | 62.3382 | 18.42 | 192 | 6.8 | 17144 | 83.77 |
| Delmar Lake | -112.055 | 63.1382 | 8.65 | 406 | 8.1 | 7885 | 82.08 |
| Denis Lake | -112.595 | 63.3542 | 7.34 | 387 | 9.2 | 6569 | 80.52 |
| Desperation Lake | -112.401 | 62.5781 | 26.04 | 244 | 21.9 | 25205 | 87.1 |
| Dessert Lake | -115.76 | 62.0993 | 7.68 | 202 | 4 | 7831 | 91.8 |
| Devore Lake | -112.902 | 62.5951 | 0.94 | 270 | 4.5 | 738 | 70.21 |
| Devreker Lake | -117.318 | 64.6627 | 13.12 | 235 | 24.8 | 12774 | 87.5 |



| | | | | | | |
|---|---|---|---|---|---|---|
| Dissension Lake | -113.499 | 63.983 | 4.97 | 336 | 5.3 | 4354 | 78.87 |
| Dodds Lake | -113.424 | 63.1327 | 4.68 | 309 | 3.9 | 4020 | 77.35 |
| Dome Lake | -113.255 | 62.7624 | 2.7 | 250 | 4.4 | 2166 | 72.22 |
| Doodad Lake | -112.755 | 62.3539 | 0.27 | 257 | 2.1 | 133 | 44.44 |
| Dorothy Lake | -112.534 | 62.4523 | 3.45 | 275 | 3 | 2744 | 71.59 |
| Doyle Lake | -109.108 | 63.0974 | 13.79 | 352 | 13.4 | 12146 | 79.26 |
| Drumlin Lake | -114.32 | 64.8287 | 36.46 | 437 | 5.4 | 34338 | 84.75 |
| Drybones Lake | -112.405 | 63.5129 | 33.07 | 411 | 10.5 | 32593 | 88.69 |
| Drygeese Lake | -114.166 | 62.734 | 3.74 | 217 | 7 | 3313 | 79.68 |
| Drymeat Lake | -112.891 | 64.2536 | 7.92 | 379 | 6.7 | 7381 | 83.84 |
| Duck Lake | -114.239 | 62.4336 | 5.38 | 155 | 6.6 | 4604 | 76.95 |
| Duckfish Lake | -114.44 | 62.6736 | 5.79 | 228 | 5.3 | 4973 | 77.37 |
| Dumas Lake | -116.301 | 66.4878 | 22.91 | 351 | 9.5 | 21601 | 84.85 |
| Dumbell Lake | -111.083 | 64.0315 | 4.15 | 433 | 5.2 | 3571 | 77.35 |
| Duncan Lake | -113.96 | 62.8705 | 68.2 | 214 | 21.4 | 70900 | 93.56 |
| Egg Lake | -114.029 | 62.4897 | 0.91 | 192 | 3.7 | 638 | 62.64 |
| Eileen Lake | -107.639 | 62.2437 | 135.71 | 369 | 9.6 | 128076 | 84.94 |
| Elk River | -105.359 | 62.2166 | 59.41 | 337 | 4.4 | 49250 | 74.62 |
| Ellington Lake | -117.32 | 65.0299 | 26.54 | 248 | 16.7 | 25216 | 85.49 |
| Ernie Lake | -102.352 | 63.2671 | 20.99 | 252 | 12.2 | 21080 | 90.38 |
| Etna Lake | -119.484 | 64.4488 | 45.73 | 356 | 3.3 | 46177 | 90.88 |
| Eyeberry Lake | -104.696 | 63.1425 | 81.6 | 201 | 5.1 | 82207 | 90.67 |
| Eyston Lake | -116.417 | 65.1823 | 23.89 | 298 | 27 | 22348 | 84.18 |
| Faber Lake | -117.297 | 63.9325 | 383.34 | 203 | 22 | 402498 | 94.5 |
| Face Lake | -110.126 | 62.3145 | 2.47 | 371 | 6.2 | 1750 | 63.56 |
| Fairbairn Lake | -111.006 | 62.2618 | 14.6 | 169 | 12.8 | 14698 | 90.62 |
| Fat Lake | -111.64 | 63.3964 | 12.85 | 412 | 7.5 | 11492 | 80.47 |
| Faulkner Lake | -112.275 | 62.1985 | 2.37 | 204 | 9.6 | 1906 | 72.57 |
| Fawn Lake | -117.529 | 62.1864 | 24.82 | 179 | 3.3 | 25619 | 92.91 |
| Fenton Lake | -112.953 | 63.0183 | 16.2 | 277 | 23.1 | 15743 | 87.47 |
| Fenwick Lake | -119.1 | 65.3622 | 3.14 | 205 | 11.1 | 2890 | 82.8 |
| Fiddlers Lake | -114.509 | 62.468 | 0.28 | 192 | 2.2 | 196 | 64.29 |
| Finger Lake | -114.357 | 62.5751 | 0.05 | 0 | 0 | 10 | 20 |
| Fishhook Lake | -115.236 | 64.0626 | 8.4 | 262 | 19.2 | 7584 | 81.31 |
| Fletcher Lake | -108.763 | 63.5923 | 164.24 | 388 | 11.6 | 164217 | 89.99 |
| Forcier Lake | -116.351 | 66.0568 | 10.1 | 371 | 9.5 | 9373 | 83.56 |
| Ford Lake | -107.409 | 63.1433 | 35.02 | 389 | 4 | 33188 | 85.29 |
| Fortune Lake | -115.183 | 64.4511 | 0.36 | 353 | 4.4 | 232 | 58.33 |
| Fox Lake | -114.417 | 62.483 | 0.52 | 199 | 2.7 | 359 | 61.54 |
| Frame Lake | -114.391 | 62.4542 | 0.85 | 186 | 3.4 | 523 | 55.29 |
| Francois Lake | -112.373 | 62.461 | 24.16 | 269 | 6.1 | 24259 | 90.36 |
| Frodsham Lake | -113.604 | 63.6462 | 23.34 | 336 | 11.8 | 23316 | 89.8 |
| Gagnon Lake | -110.45 | 62.0308 | 22.99 | 317 | 19.8 | 19590 | 76.69 |
| Gale Lake | -115.268 | 63.9338 | 1.1 | 277 | 7.6 | 863 | 70.91 |
| Gamey Lake | -115.204 | 64.1352 | 0.54 | 328 | 5.5 | 370 | 61.11 |
| Gar Lake | -114.373 | 62.5212 | 0.28 | 182 | 2.9 | 188 | 60.71 |
| Garde Lake | -106.268 | 62.832 | 104.05 | 0 | 0 | 99319 | 85.91 |
| Gardenia Lake | -105.893 | 62.0199 | 40.35 | 361 | 5 | 36955 | 82.43 |
| Georic Lake | -112.984 | 63.1825 | 0.6 | 324 | 6.2 | 398 | 60 |
| Germaine Lake | -114.609 | 63.2969 | 22.84 | 265 | 16.2 | 20542 | 80.95 |
| Ghost Lake | -115.147 | 63.8504 | 62.93 | 275 | 34.9 | 59083 | 84.49 |
| Giauque Lake | -113.831 | 63.1825 | 16.46 | 252 | 22.8 | 15472 | 84.57 |
| Glowworm Lake | -109.24 | 64.6365 | 102.5 | 412 | 20.4 | 102052 | 89.61 |
| Gold Lake | -107.949 | 64.7369 | 2.87 | 325 | 14.4 | 2113 | 66.2 |
| Goodspeed Lake | -109.465 | 63.098 | 57.92 | 319 | 24.2 | 53085 | 82.49 |
| Goodwin Lake | -114.088 | 63.0453 | 2.82 | 248 | 7.4 | 2484 | 79.43 |
| Gordon Lake | -113.201 | 63.0668 | 167.16 | 284 | 11.5 | 157999 | 85.07 |
| Grace 2 Lake | -112.564 | 62.1628 | 3.7 | 218 | 5.7 | 3093 | 75.14 |
| Grace Lake | -114.448 | 62.4188 | 0.64 | 172 | 3.4 | 378 | 53.12 |
| Graham Lake | -113.807 | 62.9008 | 16.37 | 216 | 17.8 | 16453 | 90.47 |
| Gras Lake | -110.448 | 64.523 | 705.63 | 404 | 9.6 | 724732 | 92.44 |
| Great Slave Lake | -113.243 | 62.2183 | 9553.89 | 148 | 59.1 | 8607738 | 53.14 |
| Greenrock Lake | -116.512 | 65.9309 | 2.8 | 394 | 7.8 | 2399 | 77.14 |
| Greyling Lake | -114.289 | 62.6827 | 0.5 | 198 | 5.9 | 274 | 50 |



| | | | | | | |
|---|---|---|---|---|---|---|
| Grizzle Bear Lake | -112.982 | 64.1998 | 21.13 | 376 | 5.6 | 20776 | 88.36 |
| Grizzly Lake | -115.574 | 64.5081 | 4.6 | 327 | 11.9 | 4108 | 80.43 |
| Grodsky Lake | -108.391 | 62.082 | 5.87 | 372 | 3.7 | 5652 | 86.54 |
| Hair Lake | -110.048 | 62.4313 | 5.24 | 185 | 16.8 | 4952 | 85.11 |
| Hanbury Lake | -105.698 | 63.5646 | 7.57 | 328 | 8 | 7295 | 86.79 |
| Handle lake | -114.397 | 62.4914 | 0.21 | 196 | 2.3 | 116 | 47.62 |
| Handley Page Lake | -116.777 | 65.9876 | 29.03 | 315 | 17.3 | 28307 | 87.77 |
| Hansen Lake | -116.748 | 65.6957 | 16.33 | 323 | 17.6 | 15363 | 84.69 |
| Harald Lake | -113.535 | 62.685 | 4.12 | 260 | 3.3 | 3060 | 66.75 |
| Hardisty Lake | -117.676 | 64.5506 | 302.59 | 186 | 26.9 | 298804 | 88.86 |
| Harrison Lake | -107.659 | 63.0102 | 4.71 | 402 | 3.9 | 3464 | 66.24 |
| Havant Lake | -115.555 | 65.8333 | 9.79 | 401 | 7.8 | 9524 | 87.33 |
| Haywood Lake | -110.504 | 63.4599 | 32.79 | 395 | 8.2 | 32691 | 89.72 |
| Healey Lake | -106.663 | 64.2964 | 153.4 | 352 | 8.2 | 141117 | 82.67 |
| Heuss Lake | -107.081 | 63.3052 | 9.63 | 360 | 6.1 | 7831 | 73.21 |
| Hidden 1 Lake | -113.682 | 62.5107 | 0.2 | 193 | 5 | 104 | 45 |
| Hidden Lake | -113.556 | 62.5531 | 12.45 | 205 | 14.7 | 12189 | 88.11 |
| Hilltop Lake | -111.041 | 63.3671 | 25.91 | 407 | 5.7 | 23714 | 82.36 |
| Hislop Lake | -116.927 | 63.5189 | 34.04 | 172 | 14.6 | 33779 | 89.28 |
| Hoare Lake | -105.131 | 63.6208 | 8.66 | 305 | 5 | 7769 | 80.72 |
| Holmason Lake | -115.024 | 63.9889 | 1.27 | 269 | 13.9 | 1065 | 75.59 |
| Homer Lake | -114.286 | 62.6641 | 0.59 | 0 | 0 | 358 | 54.24 |
| Hottah Lake | -118.484 | 65.0678 | 842.81 | 175 | 6.7 | 875662 | 93.45 |
| Howard Lake | -105.97 | 62.213 | 186.86 | 345 | 4 | 172970 | 83.31 |
| Huff Lake | -107.163 | 62.2876 | 7.96 | 387 | 4.9 | 6599 | 74.62 |
| Hump Lake | -116.552 | 63.586 | 2.72 | 191 | 8.2 | 2111 | 69.85 |
| Humpy Lake | -113.435 | 64.6678 | 23.77 | 402 | 8.2 | 23502 | 88.98 |
| Hunter Lake | -113.371 | 64.1024 | 9.9 | 362 | 8.9 | 9181 | 83.43 |
| Indian Hill Lake | -110.736 | 63.222 | 90.09 | 380 | 4.1 | 82097 | 82.02 |
| Indian Mountain Lake | -111.004 | 63.1267 | 23.55 | 387 | 15.6 | 21855 | 83.52 |
| Indin Lake | -115.151 | 64.2435 | 156.43 | 253 | 45.2 | 150631 | 86.66 |
| Inglis Lake | -115.164 | 63.1693 | 16.49 | 201 | 17.2 | 13740 | 75.02 |
| Ingray Lake | -116.171 | 64.2701 | 139.71 | 241 | 62.9 | 142207 | 91.6 |
| Irritation Lake | -115.264 | 65.0587 | 4.3 | 382 | 7.1 | 4019 | 84.19 |
| Isabella Lake | -117.697 | 64.8136 | 59.6 | 186 | 26.9 | 60442 | 91.19 |
| Island Lake | -114.1 | 62.4914 | 0.9 | 205 | 3.3 | 563 | 56.67 |
| Itchen Lake | -112.827 | 65.5184 | 137.71 | 400 | 23.1 | 144870 | 94.68 |
| Jackfish Lake | -114.392 | 62.4666 | 0.47 | 182 | 5 | 376 | 72.34 |
| Jackson Lake | -114.305 | 62.5872 | 0.92 | 182 | 5.2 | 654 | 64.13 |
| James Lake | -116.439 | 63.0095 | 19.34 | 147 | 15.8 | 17730 | 82.52 |
| Jennejohn Lake | -113.747 | 62.4206 | 16.98 | 196 | 4.6 | 15192 | 80.51 |
| Jim Lake | -104.579 | 62.4087 | 20.33 | 282 | 5.7 | 20581 | 91.1 |
| Joe Lake | -114.387 | 62.483 | 0.1 | 0 | 0 | 45 | 40 |
| Johnston Lake | -114.2 | 62.9965 | 5.34 | 213 | 10.6 | 4969 | 83.71 |
| Jolly Lake | -111.94 | 64.1417 | 73.43 | 403 | 9.8 | 75697 | 92.78 |
| Jones Lake | -108.377 | 62.3131 | 4.53 | 367 | 7.7 | 3984 | 79.25 |
| Kam Lake | -114.406 | 62.4205 | 2.12 | 163 | 4.7 | 1770 | 75 |
| Kamilukuak Lake | -102.005 | 62.4711 | 0.34 | 247 | 21.1 | 235 | 61.76 |
| Keskarrah Lake | -115.25 | 66.0464 | 18.73 | 399 | 8.3 | 17792 | 85.37 |
| King Lake | -110.762 | 63.7752 | 13.04 | 402 | 5.7 | 12294 | 84.82 |
| Kirk Lake | -109.066 | 63.7188 | 64.1 | 389 | 5.7 | 64275 | 90.25 |
| Kog lake | -114.396 | 62.4048 | 0.63 | 170 | 3.3 | 349 | 49.21 |
| Koropchuk Lake | -116.767 | 64.1512 | 27.7 | 241 | 9.4 | 23818 | 77.4 |
| Kway Cha Lake | -118.552 | 65.4413 | 23.96 | 166 | 2.8 | 23902 | 89.69 |
| La Loche Lakes | -110.877 | 62.006 | 2.65 | 301 | 6.3 | 2423 | 82.26 |
| Lac Avril | -115.292 | 63.9542 | 1.35 | 253 | 9.9 | 1123 | 74.81 |
| Lac de Charloit | -107.976 | 63.8105 | 102.31 | 375 | 13.5 | 104407 | 91.85 |
| Lac du Bois | -105.76 | 63.6146 | 13.71 | 343 | 3.8 | 12312 | 80.82 |
| Lac Grandin | -119.064 | 63.9802 | 244.36 | 319 | 6.1 | 263747 | 97.14 |
| Lac la Martre | -117.961 | 63.3195 | 1676.65 | 249 | 10 | 1815957 | 97.34 |
| Lac la Prise | -108.722 | 63.062 | 30.61 | 375 | 9.9 | 29779 | 87.55 |
| Lac Levis | -117.951 | 62.6347 | 53.25 | 265 | 3.7 | 56088 | 94.8 |
| Lac Malfait | -117.988 | 64.6284 | 38.05 | 174 | 4.8 | 38337 | 90.67 |
| Lac Nez Croche | -111.403 | 63.2502 | 28.59 | 396 | 9.7 | 27563 | 86.78 |





| | | | | | | |
|---|---|---|---|---|---|---|
| Lac Tachs | -119.987 | 64.0127 | 124.55 | 308 | 13.5 | 126349 | 91.3 |
| Lac Tate dOurs | -110.572 | 63.3577 | 62.23 | 385 | 10 | 63718 | 92.16 |
| Lake of the Enemy | -110.238 | 63.7792 | 135.33 | 396 | 12.2 | 134187 | 89.06 |
| Lake Providence | -112.065 | 64.7582 | 102.58 | 358 | 28 | 103207 | 90.55 |
| Lamoureux Lake | -113.682 | 62.9108 | 4.11 | 270 | 6.1 | 3551 | 77.86 |
| Landing Lake | -114.408 | 62.56 | 1.28 | 202 | 2.6 | 861 | 60.16 |
| Languish Lake | -112.904 | 62.7684 | 11.24 | 307 | 4.4 | 10597 | 84.88 |
| Larocque Lake | -107.707 | 63.0496 | 1.5 | 396 | 2.5 | 1108 | 66.67 |
| Lastfire Lake | -113.029 | 64.5342 | 6.13 | 357 | 6.4 | 5919 | 86.95 |
| Laurie Lake | -115.208 | 64.4817 | 0.47 | 358 | 5 | 330 | 63.83 |
| Lausen Lake | -109.744 | 62.5875 | 7.24 | 186 | 15.3 | 6648 | 82.6 |
| Leonforte Lake | -119.654 | 64.5848 | 10.65 | 392 | 6.1 | 10916 | 92.21 |
| Likely Lake | -114.311 | 62.6466 | 0.39 | 204 | 4.9 | 130 | 30.77 |
| Little Crapeau Lake | -116.518 | 64.8211 | 123.5 | 275 | 21.4 | 118280 | 85.98 |
| Little Forehead Lake | -113.281 | 64.7824 | 18.2 | 401 | 20 | 16804 | 83.08 |
| Little Lake | -113.96 | 62.5467 | 0.14 | 217 | 2.7 | 92 | 57.14 |
| Logie Lake | -105.759 | 62.1353 | 4.23 | 357 | 2.8 | 3454 | 73.52 |
| Long Lake | -114.442 | 62.4773 | 1.16 | 196 | 2.1 | 788 | 61.21 |
| Long Legs Lake | -113.773 | 64.7613 | 16.7 | 417 | 5.6 | 14850 | 80.06 |
| Longtom Lake | -117.834 | 65.1715 | 43.93 | 189 | 19.3 | 43869 | 89.87 |
| Lou Lake | -116.782 | 63.5682 | 1.82 | 193 | 13.8 | 1468 | 72.53 |
| Love Lake | -114.759 | 62.995 | 4.43 | 246 | 5.9 | 3172 | 64.33 |
| Lynx Lake | -106.285 | 62.4099 | 295.27 | 344 | 10.1 | 282676 | 86.16 |
| Mac Lake | -113.468 | 63.0753 | 5.09 | 312 | 5.3 | 3993 | 70.53 |
| MacKay Lake | -111.012 | 63.9233 | 972.33 | 390 | 20.7 | 1002573 | 92.8 |
| MacLellan Lake | -110.037 | 63.2324 | 23.2 | 383 | 13.9 | 20923 | 81.16 |
| MacNaughton Lake | -115.314 | 63.7135 | 0.37 | 296 | 3.7 | 201 | 48.65 |
| Mad Lake | -112.749 | 62.1169 | 1.16 | 222 | 4.1 | 958 | 74.14 |
| Madeline Lake | -114.081 | 62.5468 | 0.93 | 170 | 8.8 | 733 | 70.97 |
| Magpie Lake | -108.877 | 62.4488 | 4.13 | 355 | 8.8 | 3700 | 80.63 |
| Magrum Lake | -108.635 | 62.0674 | 4.97 | 373 | 7.1 | 4492 | 81.29 |
| Malley Lake | -108.087 | 63.5601 | 30.34 | 395 | 7.5 | 25931 | 76.93 |
| Mann Lake | -112.797 | 62.3496 | 1.26 | 244 | 2.2 | 958 | 68.25 |
| Mantic Lake | -104.457 | 62.3336 | 59.48 | 293 | 5.4 | 58030 | 87.81 |
| Margaret Lake | -117.128 | 64.507 | 102.43 | 203 | 32.2 | 101844 | 89.47 |
| Marian Lake | -116.203 | 62.9243 | 236.84 | 147 | 15.8 | 251024 | 95.22 |
| Martin Lake | -114.439 | 62.5313 | 3.09 | 197 | 3.2 | 2163 | 63.11 |
| Mary Frances Lake | -106.245 | 63.3043 | 149.45 | 363 | 8 | 143654 | 86.51 |
| Mary Lake | -103.54 | 62.3855 | 164.8 | 294 | 20.6 | 170394 | 92.96 |
| Mattberry Lake | -115.891 | 64.1132 | 82.24 | 235 | 38.6 | 80356 | 87.94 |
| Matthews Lake | -111.245 | 64.0696 | 10.35 | 422 | 6.3 | 9582 | 83.29 |
| Max Ward Lake | -113.708 | 65.4787 | 11.65 | 367 | 8.5 | 11912 | 92.02 |
| Maze Lake | -105.944 | 63.8946 | 16.16 | 350 | 2.9 | 12516 | 69.68 |
| Mazenod Lake | -117.014 | 63.7003 | 36.3 | 199 | 14 | 34834 | 86.36 |
| McCrea Lake | -112.572 | 63.5556 | 16.07 | 404 | 12 | 15065 | 84.38 |
| McIntosh Lake | -114.901 | 65.7574 | 3.71 | 436 | 5 | 3398 | 82.48 |
| McKee Lake | -110.043 | 62.3492 | 2.97 | 353 | 6.1 | 2387 | 72.39 |
| McKinlay Lake | -111.541 | 62.8749 | 26.65 | 365 | 8.1 | 25518 | 86.19 |
| McKinnon Lake | -108.497 | 62.0601 | 10.03 | 370 | 5.6 | 9324 | 83.65 |
| McLellan Lake | -117.958 | 63.8428 | 9.55 | 0 | 0 | 9510 | 89.63 |
| McPhee Lake | -113.052 | 63.0264 | 1.91 | 312 | 7.2 | 1323 | 62.3 |
| McTavish Arm | -117.958 | 65.4491 | 189.7 | 175 | 29.6 | 188745 | 89.55 |
| Meander Lake | -112.149 | 62.5774 | 10.48 | 311 | 10.7 | 9539 | 81.97 |
| Meg Lake | -114.383 | 62.416 | 0.09 | 0 | 0 | 45 | 44.44 |
| Meridian Lake | -109.43 | 62.6042 | 37.1 | 201 | 26.3 | 38923 | 94.42 |
| Merl Lake | -112.655 | 62.4007 | 0.7 | 260 | 3.5 | 550 | 70 |
| Mesa Lake | -115.147 | 64.8268 | 36.55 | 365 | 12.4 | 34701 | 85.44 |
| Messina Lake | -119.526 | 64.1837 | 18.27 | 371 | 7 | 18633 | 91.79 |
| Methane Lake | -114.174 | 62.4838 | 0.98 | 180 | 3.8 | 684 | 63.27 |
| Michel Lake | -114.141 | 62.881 | 3.17 | 229 | 6.1 | 2633 | 74.76 |
| Milner lake | -114.341 | 62.5923 | 0.41 | 212 | 2.3 | 182 | 39.02 |
| Misty Lake | -109.785 | 63.067 | 9.86 | 321 | 12.9 | 9180 | 83.77 |
| Moberly Lake | -114.315 | 63.0166 | 12.75 | 225 | 20.8 | 10564 | 74.59 |
| Mohawk Lake | -112.115 | 64.0222 | 20.01 | 438 | 4.6 | 18774 | 84.46 |



| Moise Lake | -114.136 | 62.3247 | 0.78 | 166 | 3.7 | 596 | 69.23 |
|---|---|---|---|---|---|---|---|
| Moose Lake | -114.089 | 62.9803 | 1.24 | 253 | 5.7 | 978 | 70.97 |
| Moraine Lake | -106.01 | 64.1074 | 80.99 | 352 | 5.3 | 75416 | 83.8 |
| Morel Lake | -113.677 | 65.6372 | 20.28 | 380 | 8 | 19808 | 87.92 |
| Morose Lake | -112.915 | 62.8252 | 12.01 | 311 | 6 | 11015 | 82.51 |
| Mosquito Lake | -103.341 | 62.5798 | 311.21 | 292 | 12.7 | 323804 | 93.54 |
| Mud Lake | -117.197 | 63.0225 | 11.2 | 0 | 0 | 9829 | 79.02 |
| Munn Lake | -109.974 | 63.6481 | 70.94 | 391 | 15.9 | 71057 | 90.01 |
| Murdock Lake | -109.431 | 63.5811 | 51.1 | 423 | 4 | 44286 | 78 |
| Murphy Lake | -109.801 | 62.1121 | 8.86 | 299 | 9.4 | 8198 | 83.3 |
| Murray Lake | -113.441 | 63.0154 | 2.56 | 292 | 3.7 | 2187 | 76.95 |
| Musclow Lake | -106.953 | 63.7879 | 4.97 | 375 | 3.2 | 4259 | 77.06 |
| Muskeg Lake | -103.64 | 62.0805 | 8.47 | 322 | 5.4 | 6939 | 73.67 |
| Naga Lake | -119.21 | 65.2199 | 5.31 | 175 | 5.7 | 5261 | 89.08 |
| Nardin Lake | -113.839 | 63.4931 | 18.14 | 341 | 11.9 | 15645 | 77.62 |
| Nelligan Lake | -105.784 | 63.3149 | 8.66 | 360 | 4.1 | 7482 | 77.71 |
| Nelson Lake | -108.111 | 62.1925 | 7.77 | 409 | 3.5 | 5840 | 67.7 |
| Newbigging Lake | -112.226 | 64.4419 | 15.36 | 444 | 8 | 15039 | 88.15 |
| Nieznany Lake | -105.175 | 62.3904 | 3.56 | 323 | 3.9 | 2966 | 75 |
| Nonacho Lake | -109.317 | 62.0843 | 104.88 | 312 | 15.3 | 104841 | 89.86 |
| NoName Lake 01 | -114.39 | 62.5522 | 0.06 | 0 | 0 | 14 | 16.67 |
| NoName Lake 02 | -108.095 | 64.4885 | 50.13 | 379 | 14.1 | 48651 | 87.35 |
| NoName Lake 03 | -109.052 | 64.3012 | 28.16 | 403 | 4.6 | 28308 | 90.48 |
| NoName Lake 04 | -109.297 | 64.3279 | 64.28 | 393 | 6.3 | 65424 | 91.6 |
| NoName Lake 05 | -108.275 | 63.7958 | 20.95 | 389 | 6.2 | 20814 | 89.4 |
| NoName Lake 06 | -112.075 | 63.6362 | 70.68 | 413 | 7.3 | 68296 | 86.97 |
| NoName Lake 07 | -110.554 | 63.5828 | 64.99 | 409 | 8.4 | 59077 | 81.81 |
| NoName Lake 08 | -114.046 | 63.3532 | 17.4 | 290 | 28.3 | 16528 | 85.52 |
| NoName Lake 09 | -107.279 | 63.3114 | 77.05 | 362 | 6.1 | 74521 | 87.05 |
| NoName Lake 10 | -106.178 | 63.1366 | 47.57 | 349 | 5.9 | 47674 | 90.2 |
| NoName Lake 11 | -117.93 | 62.9845 | 45.5 | 267 | 3.1 | 46436 | 91.85 |
| NoName Lake 12 | -102.796 | 63.0017 | 88.98 | 263 | 4.3 | 86218 | 87.21 |
| NoName Lake 13 | -111.917 | 62.9299 | 40.08 | 390 | 13.8 | 39749 | 89.25 |
| NoName Lake 14 | -102.003 | 62.6587 | 34.76 | 246 | 8.7 | 31634 | 81.9 |
| NoName Lake 15 | -114.246 | 62.7732 | 24.89 | 195 | 21.1 | 20640 | 74.65 |
| NoName Lake 16 | -113.936 | 62.5755 | 38.89 | 166 | 20.8 | 36375 | 84.19 |
| NoName Lake 17 | -114.194 | 62.5985 | 36.73 | 152 | 24.4 | 36116 | 88.48 |
| NoName Lake 18 | -107.594 | 62.4351 | 48.91 | 362 | 8.7 | 46675 | 85.89 |
| NoName Lake 19 | -103.28 | 62.2774 | 62.81 | 285 | 8.5 | 60275 | 86.29 |
| NoName Lake 20 | -114.47 | 62.6374 | 1.1 | 215 | 4 | 661 | 53.64 |
| NoName Lake 21 | -114.421 | 62.5109 | 1.33 | 201 | 1.6 | 919 | 62.41 |
| NoName Lake 22 | -114.231 | 62.4853 | 3.21 | 169 | 5.3 | 2700 | 75.7 |
| NoName Lake 23 | -114.177 | 62.4598 | 1.39 | 164 | 2.9 | 1069 | 69.06 |
| NoName Lake 24 | -114.628 | 62.428 | 1.26 | 171 | 2.6 | 891 | 63.49 |
| NoName Lake 25 | -114.472 | 62.5003 | 0.21 | 204 | 1.9 | 131 | 57.14 |
| NoName Lake 26 | -109.632 | 65.0632 | 208.76 | 427 | 15.1 | 208919 | 89.95 |
| NoName Lake 27 | -110.876 | 64.792 | 109.48 | 430 | 8.2 | 108844 | 89.48 |
| NoName Lake 28 | -115.883 | 63.7155 | 139.74 | 208 | 2 | 129795 | 83.48 |
| NoName Lake 29 | -112.318 | 63.4001 | 106.08 | 397 | 14.5 | 100296 | 85.1 |
| NoName Lake 30 | -102.65 | 62.5706 | 194.39 | 255 | 11.7 | 192351 | 89.06 |
| NoName Lake 31 | -114.871 | 65.2597 | 25.88 | 449 | 5.6 | 25720 | 89.45 |
| NoName Lake 32 | -115.923 | 65.0306 | 54.52 | 348 | 18.5 | 52898 | 87.33 |
| NoName Lake 33 | -109.082 | 65.024 | 68.24 | 399 | 10.9 | 71432 | 94.21 |
| NoName Lake 34 | -108.844 | 64.9761 | 36.5 | 380 | 7.7 | 38642 | 95.29 |
| NoName Lake 35 | -109.044 | 64.9043 | 19.27 | 417 | 12.8 | 19406 | 90.66 |
| NoName Lake 36 | -108.66 | 64.9469 | 44.71 | 382 | 6.7 | 45389 | 91.37 |
| NoName Lake 37 | -109 | 64.7693 | 65.26 | 397 | 9.9 | 64212 | 88.55 |
| Noyes Lake | -105.901 | 62.5395 | 24.79 | 346 | 4.3 | 24435 | 88.71 |
| Octopus Lake | -114.449 | 62.3737 | 0.77 | 158 | 1.8 | 453 | 53.25 |
| Odjick Lake | -113.917 | 65.516 | 31.55 | 358 | 28 | 32155 | 91.73 |
| Old Canoe Lake | -111.453 | 63.443 | 61.76 | 421 | 12.5 | 57663 | 84.03 |
| Olson Lake | -105.277 | 62.9121 | 7.97 | 337 | 4 | 7755 | 87.58 |
| One Arm Lake | -114.342 | 62.5458 | 0.12 | 183 | 3 | 37 | 25 |
| Orkney Lake | -113.182 | 64.1307 | 5.96 | 385 | 5 | 5583 | 84.23 |





| Oro Lake | -114.333 | 62.6283 | 0.46 | 219 | 3.5 | 231 | 45.65 |
|---|---|---|---|---|---|---|---|
| Ortona Lake | -119.222 | 64.7705 | 14.3 | 634 | 6.2 | 14279 | 89.86 |
| Outram Lakes | -109.433 | 64.0362 | 48.87 | 367 | 9.2 | 49430 | 91.04 |
| Papanakies Lake | -110.338 | 63.2306 | 23.3 | 406 | 5.7 | 22211 | 85.79 |
| Parent Lake | -114.381 | 65.2658 | 50.14 | 376 | 14.6 | 49992 | 89.73 |
| Pate Lake | -114.206 | 64.4237 | 12.02 | 382 | 7.4 | 11154 | 83.53 |
| Payne Lake | -112.068 | 62.8293 | 9.74 | 362 | 12.1 | 8904 | 82.24 |
| Peaceful Lake | -113.505 | 62.9932 | 2.46 | 283 | 4.2 | 2119 | 77.64 |
| Pellatt Lake | -109.777 | 64.9606 | 40.31 | 427 | 15.1 | 38887 | 86.83 |
| Pelonquin Lake | -111.225 | 65.3221 | 21.08 | 493 | 7 | 21342 | 91.13 |
| Peninsula Lake | -113.364 | 62.5234 | 0.84 | 230 | 4.2 | 501 | 53.57 |
| Perlson Lake | -111.92 | 63.1328 | 24.87 | 394 | 9.1 | 24483 | 88.58 |
| Phoenix Lake | -113.339 | 63.7636 | 27.15 | 344 | 13.9 | 24178 | 80.15 |
| Pickerel Lake | -113.488 | 62.4943 | 1.57 | 209 | 6.8 | 1100 | 63.06 |
| Pink Lake | -113.018 | 62.6731 | 3.44 | 263 | 7.4 | 2883 | 75.29 |
| Plex Lake | -110.785 | 63.1137 | 2.21 | 389 | 5.8 | 2036 | 82.81 |
| Point Lake | -113.091 | 65.2602 | 626.84 | 358 | 28 | 644500 | 92.47 |
| Pollock Lake | -115.808 | 63.3195 | 4.06 | 198 | 9.3 | 3165 | 70.2 |
| Pontoon Lake | -114.003 | 62.5418 | 3.36 | 195 | 7.2 | 3064 | 82.14 |
| Porphyry Lake | -113.403 | 64.0488 | 3.53 | 345 | 6.9 | 3156 | 80.45 |
| Prang Lake | -112.501 | 63.8773 | 14.34 | 425 | 7.9 | 13060 | 81.94 |
| Preg Lake | -114.081 | 62.4527 | 0.17 | 173 | 3.3 | 76 | 41.18 |
| Prestige Lake | -113.645 | 62.9615 | 8.25 | 270 | 6.8 | 7802 | 85.09 |
| Price Lake | -108.158 | 62.0349 | 8.59 | 375 | 7 | 8341 | 87.43 |
| Ptarmigan Lake | -107.429 | 63.5903 | 82.73 | 352 | 12.7 | 82480 | 89.71 |
| Pud lake | -114.383 | 62.4316 | 0.17 | 181 | 2.7 | 96 | 52.94 |
| Rabbit Lake | -116.849 | 63.4668 | 11.96 | 172 | 12.4 | 12441 | 93.65 |
| Raccoon Lake | -117.692 | 62.87 | 43.93 | 287 | 1 | 45951 | 94.15 |
| Radford Lake | -105.576 | 63.3944 | 41.05 | 341 | 4.5 | 39226 | 85.99 |
| Rae Lake | -117.321 | 64.1656 | 201.35 | 200 | 28.3 | 198474 | 88.72 |
| Range Lake | -114.423 | 62.4473 | 0.21 | 188 | 2.9 | 110 | 47.62 |
| Ranji Lake | -115.09 | 64.1015 | 15.29 | 260 | 29.8 | 15364 | 90.45 |
| Rater Lake | -114.368 | 62.5537 | 0.2 | 182 | 3.8 | 51 | 25 |
| Rawalpindi Lake | -114.623 | 65.0285 | 88.37 | 415 | 6.7 | 87345 | 88.96 |
| Rebesca Lake | -116.373 | 64.5352 | 65.24 | 252 | 36.5 | 66388 | 91.46 |
| Recluse Lake | -114.015 | 66.0421 | 0.71 | 376 | 13.5 | 522 | 66.2 |
| Redout Lake | -113.016 | 62.7403 | 8.38 | 293 | 7.4 | 7423 | 79.71 |
| Redrock Lake | -114.165 | 65.4776 | 83.25 | 358 | 28 | 82622 | 89.32 |
| Reid Lake | -109.959 | 63.7626 | 40.8 | 401 | 8.9 | 39960 | 88.14 |
| Reindeer Lake | -113.583 | 63.8865 | 50.25 | 338 | 10.5 | 49658 | 88.94 |
| Rib Lake | -114.177 | 62.3445 | 0.58 | 166 | 4 | 430 | 67.24 |
| River Lake | -114.091 | 62.5945 | 4.96 | 166 | 20.8 | 4495 | 81.65 |
| Robb Lake | -116.021 | 65.3709 | 16.95 | 356 | 15.4 | 17118 | 90.91 |
| Robert Lake | -109.357 | 62.3803 | 3.52 | 331 | 11.8 | 2948 | 75.28 |
| Rodrigues Lake | -115.633 | 64.7871 | 5.55 | 296 | 11.5 | 5399 | 87.57 |
| Rolfe Lake | -111.725 | 63.0835 | 55.57 | 402 | 8.2 | 54011 | 87.48 |
| Rome Lake | -118.342 | 64.3155 | 22.23 | 402 | 3.3 | 21396 | 86.64 |
| Ross Lake | -113.26 | 62.6815 | 15.7 | 254 | 7 | 14207 | 81.46 |
| Roulante Lake | -113.748 | 64.5571 | 20.52 | 420 | 11.4 | 18449 | 80.9 |
| Roundrock Lake | -113.404 | 64.3891 | 29.74 | 342 | 23.9 | 29970 | 90.69 |
| Rupp Lake | -112.264 | 63.8287 | 6.37 | 442 | 4.8 | 5234 | 73.94 |
| Russell Lake | -115.75 | 63.0373 | 177.03 | 147 | 15.8 | 171240 | 86.93 |
| Ryan Lake | -114.372 | 62.5871 | 1.06 | 220 | 2.6 | 879 | 74.53 |
| Samandr Lake | -115.384 | 65.9782 | 59.36 | 422 | 10.8 | 59573 | 90.11 |
| Sandy Lake | -113.077 | 64.1536 | 4.18 | 382 | 6.6 | 3763 | 80.86 |
| Sarah Lake | -117.147 | 63.7832 | 66.45 | 187 | 17.3 | 66049 | 89.45 |
| Savannah Lake | -108.911 | 64.4309 | 29.02 | 393 | 6.6 | 27969 | 86.73 |
| Savoy Lake | -115.436 | 64.4286 | 1 | 330 | 7.7 | 773 | 70 |
| Schist Lakes | -109.913 | 62.3707 | 0.95 | 353 | 5.2 | 422 | 40 |
| Schwerdt Lake | -115.268 | 64.3828 | 0.1 | 0 | 0 | 37 | 30 |
| Scott Lake | -113.572 | 62.652 | 2.62 | 246 | 5.6 | 1883 | 64.5 |
| Scotty Lake | -112.989 | 63.4673 | 1.67 | 408 | 8.6 | 1359 | 73.05 |
| Seahorse Lake | -111.229 | 64.3086 | 20.41 | 420 | 5.4 | 20629 | 90.98 |
| Seal Lake | -108.95 | 64.6326 | 83.6 | 403 | 13.1 | 79887 | 86 |





| Second Lake | -117.426 | 62.1259 | 3.27 | 179 | 5.4 | 3122 | 85.93 |
|---|---|---|---|---|---|---|---|
| Self Lake | -117.274 | 65.2949 | 21.46 | 292 | 14.3 | 20770 | 87.09 |
| Shadow Lake | -114.35 | 62.5662 | 0.06 | 0 | 0 | 13 | 16.67 |
| Shamrock Lake | -115.012 | 64.7763 | 17.82 | 367 | 8.9 | 17503 | 88.38 |
| Shaw Lake | -112.765 | 64.6106 | 4.99 | 392 | 6.2 | 4290 | 77.35 |
| Short Point Lake | -114.224 | 62.7569 | 5.69 | 195 | 21.1 | 5339 | 84.53 |
| Sid Lake | -103.986 | 62.2425 | 289.48 | 296 | 12 | 304865 | 94.78 |
| Sifton Lake | -106.36 | 63.7027 | 90.93 | 355 | 6.6 | 75978 | 75.2 |
| Simon Lake | -117.318 | 65.5421 | 6.98 | 270 | 11.8 | 6211 | 80.09 |
| Singing Lake | -112.925 | 64.3162 | 13.32 | 370 | 9 | 12863 | 86.79 |
| Sled Lake | -106.821 | 62.1265 | 23.5 | 380 | 6.4 | 21841 | 83.66 |
| Sleepy Dragon Lake | -112.909 | 62.9194 | 5.21 | 331 | 5.7 | 4748 | 81.96 |
| Slemon Lake | -116.033 | 63.2081 | 44.91 | 138 | 14.7 | 44378 | 88.82 |
| Small Lake | -113.826 | 62.5185 | 0.74 | 193 | 5.1 | 547 | 66.22 |
| Smart Lake | -106.822 | 63.4912 | 112.35 | 356 | 11.3 | 103542 | 82.88 |
| Smoky Lake | -116.495 | 65.9003 | 2.45 | 369 | 9.8 | 2178 | 80 |
| Snelgrove Lake | -105.615 | 62.3356 | 7.59 | 348 | 4.4 | 6599 | 78.26 |
| Sophia Lake | -114.121 | 62.9357 | 3.64 | 248 | 7.2 | 2327 | 57.42 |
| Sosan Lake | -111.95 | 63.2369 | 5.26 | 432 | 4.6 | 4140 | 70.91 |
| Sparrow Lake | -113.648 | 62.6144 | 12.58 | 237 | 6.4 | 12077 | 86.41 |
| Spencer Lake | -112.462 | 63.1573 | 13.2 | 375 | 11.8 | 13060 | 88.86 |
| Sphinx Lake | -115.366 | 64.4645 | 1.56 | 345 | 5.7 | 1351 | 78.21 |
| Spider Lake | -115.145 | 64.5067 | 16.58 | 341 | 11.5 | 13690 | 74.31 |
| Sproule Lake | -113.478 | 62.7444 | 1.6 | 274 | 2.4 | 1022 | 57.5 |
| Spruce Island Lake | -110.427 | 62.4009 | 1.66 | 171 | 10.3 | 1419 | 77.11 |
| Staple Lake | -114.033 | 62.729 | 1.27 | 247 | 3.1 | 822 | 58.27 |
| Starfish Lake | -111.61 | 64.3321 | 21.29 | 403 | 6.3 | 21575 | 91.22 |
| Starvation Lake | -112.731 | 64.8988 | 37.76 | 400 | 9.1 | 38344 | 91.39 |
| Steel Lake | -104.593 | 63.7203 | 8.98 | 159 | 8.6 | 8562 | 85.86 |
| Sterlet Lake | -109.496 | 64.7214 | 44.66 | 426 | 15.1 | 41175 | 82.89 |
| Street Lake | -105.317 | 63.4127 | 15.43 | 326 | 2.8 | 13352 | 77.9 |
| Sunken Lake | -110.233 | 62.9846 | 1.7 | 259 | 19.1 | 1460 | 77.06 |
| Suse Lake | -112.966 | 63.1386 | 2.36 | 341 | 4.8 | 1975 | 75.42 |
| Sussex Lake | -108.328 | 64.4388 | 14.31 | 379 | 22 | 13522 | 85.05 |
| Tanco Lake | -112.223 | 62.4201 | 4.27 | 272 | 5.7 | 2945 | 62.06 |
| Tarantula Lake | -107.95 | 64.521 | 39.79 | 373 | 7.7 | 38615 | 87.33 |
| Taylor Lake | -108.664 | 63.7853 | 33.13 | 389 | 10.2 | 32402 | 88.02 |
| Tayonton Lake | -116.544 | 63.2112 | 23.03 | 150 | 3.6 | 12081 | 47.2 |
| Tent Lake | -107.957 | 62.4281 | 72.1 | 346 | 12.7 | 67843 | 84.69 |
| Terry Lake | -113.31 | 62.511 | 4.28 | 226 | 3.5 | 2570 | 53.97 |
| The Nine Lakes | -114.043 | 63.4579 | 1.59 | 324 | 5.8 | 1194 | 67.3 |
| Thetis Lake | -113.275 | 63.7214 | 29.11 | 351 | 9.9 | 28312 | 87.36 |
| Thistlethwaite Lake | -113.627 | 63.1591 | 44.27 | 252 | 22.8 | 42541 | 86.49 |
| Thomas Lake | -119.187 | 65.1207 | 3.81 | 241 | 3.8 | 3568 | 83.99 |
| Thompson Lake | -113.5 | 62.6137 | 2.81 | 252 | 2.9 | 2381 | 76.16 |
| Thonokied Lake | -109.628 | 64.3849 | 129.5 | 394 | 18.3 | 127553 | 88.55 |
| Timberhill Lake | -106.655 | 62.37 | 16.55 | 363 | 6.3 | 14940 | 81.27 |
| Toad Lake | -111.746 | 62.7272 | 5.91 | 362 | 4.8 | 4883 | 74.28 |
| Tonggot Lake | -119.697 | 63.9928 | 18.55 | 312 | 4.3 | 15529 | 75.36 |
| Toopon Lake | -110.439 | 62.3529 | 1.51 | 176 | 9.1 | 1372 | 81.46 |
| Torrie Lake | -116.926 | 66.2355 | 0.52 | 310 | 7.7 | 430 | 75 |
| Toura Lake | -108.568 | 62.8338 | 0.79 | 374 | 5 | 507 | 58.23 |
| Trapper lake | -114.363 | 62.5266 | 0.31 | 182 | 3.1 | 137 | 38.71 |
| Trout Lake | -114.364 | 62.7997 | 2.82 | 204 | 10.3 | 2400 | 76.6 |
| Truce Lake | -114.886 | 64.5323 | 28.65 | 343 | 8.4 | 27814 | 87.36 |
| Trumper Lake | -117.582 | 63.5949 | 4.96 | 0 | 0 | 4201 | 76.21 |
| Tsan Lake | -112.937 | 64.0169 | 12.79 | 374 | 5.6 | 11773 | 82.88 |
| Tuchay Lake | -119.163 | 65.2513 | 31.72 | 172 | 11.1 | 31222 | 88.56 |
| Tuche Lake | -117.317 | 64.3356 | 14.78 | 200 | 15.8 | 14489 | 88.23 |
| Tumi Lake | -116.794 | 63.4535 | 6.07 | 165 | 7.7 | 5620 | 83.36 |
| Tyrrell Lake | -105.498 | 63.1246 | 227.09 | 318 | 9.8 | 220532 | 87.4 |
| Uhlman Lake | -116.799 | 66.1321 | 9.3 | 314 | 12 | 8538 | 82.58 |
| Upper Pensive Lake | -113.393 | 62.7247 | 3.28 | 246 | 5 | 2718 | 74.7 |
| Upper Ross Lake | -113.153 | 62.7296 | 9.22 | 254 | 7 | 8451 | 82.54 |





| Ursula Lake | -110.459 | 64.8159 | 22.95 | 453 | 6.7 | 22803 | 89.41 |
|---|---|---|---|---|---|---|---|
| Vaillant Lake | -114.51 | 66.2053 | 2.46 | 329 | 11.2 | 2230 | 81.71 |
| Van Lake | -113.077 | 63.3649 | 4.48 | 340 | 11.5 | 3780 | 75.89 |
| Vee Lake | -114.35 | 62.5555 | 0.7 | 178 | 4.4 | 393 | 50 |
| Victory Lake | -113.077 | 62.6708 | 10.37 | 252 | 4.2 | 9373 | 81.39 |
| Vital Lake | -114.438 | 62.601 | 1.49 | 194 | 5.3 | 1092 | 65.77 |
| Waite Lake | -113.322 | 62.8342 | 7.62 | 290 | 3.7 | 5495 | 64.96 |
| Wallie Lake | -113.951 | 63.1343 | 0.2 | 285 | 3.4 | 98 | 45 |
| Walmsley Lake | -108.493 | 63.4197 | 231.36 | 378 | 25.8 | 233258 | 90.74 |
| Walsh Lake | -114.281 | 62.5829 | 9.17 | 174 | 9.6 | 8030 | 78.84 |
| Webb Lake | -113.125 | 62.8492 | 3.62 | 314 | 4.6 | 2650 | 65.75 |
| Wecho Lake | -113.812 | 63.9602 | 102.43 | 351 | 16.2 | 97862 | 85.99 |
| Wedge Lake | -113.69 | 62.8632 | 9.87 | 260 | 7.9 | 8665 | 79.03 |
| White Quartz Lake | -108.383 | 62.6897 | 2.6 | 380 | 6.1 | 2089 | 72.31 |
| Whitefish Lake | -106.802 | 62.6983 | 331.46 | 350 | 11.6 | 326891 | 88.68 |
| Whitewolf Lake | -113.919 | 64.9647 | 52.93 | 419 | 10.1 | 47837 | 81.33 |
| Willow Lake | -114.215 | 62.3617 | 0.9 | 162 | 5.5 | 658 | 65.56 |
| Windflower Lake | -118.517 | 62.8653 | 36.65 | 256 | 3.7 | 38202 | 93.81 |
| Windy Lake | -109.928 | 64.9443 | 8.61 | 440 | 7.4 | 8000 | 83.62 |
| Winter Lake | -112.943 | 64.4877 | 45.27 | 346 | 11.6 | 46846 | 93.09 |
| Wolverine Lake | -111.38 | 63.2084 | 23.42 | 396 | 9.7 | 23074 | 88.68 |
| Wonnacott Lake | -116.686 | 63.7158 | 1.62 | 0 | 0 | 1028 | 57.41 |
| Woyna Lake | -112.985 | 62.4704 | 2.52 | 241 | 6 | 2226 | 79.37 |
| Wylie Lake | -117.011 | 65.6689 | 15.66 | 322 | 9.4 | 14391 | 82.69 |
| Yamba Lake | -111.376 | 64.9531 | 305.28 | 403 | 16.7 | 310925 | 91.66 |
| Yanik Lake | -118.631 | 65.3664 | 8.1 | 230 | 5.2 | 8157 | 90.37 |
| Zebulon Lake | -117.853 | 65.0521 | 56.04 | 184 | 15.4 | 56562 | 90.85 |
| Zigzag Lake | -113.035 | 62.3407 | 5.11 | 200 | 10.9 | 4239 | 74.76 |
| Zinto Lake | -116.396 | 64.1152 | 52.42 | 242 | 27.9 | 48637 | 83.5 |
| Zipper Lake | -112.522 | 63.7092 | 3.81 | 426 | 6.5 | 3055 | 72.18 |
| Zucker Lake | -106.799 | 62.9326 | 53.17 | 373 | 4.2 | 47658 | 80.67 |

615