# Peer review of "Lake Surface Temperature Dataset in the North Slave Region Retrieved from Landsat Satellite Series - 1984 to 2021"

_Earth System Science Data, 2022_

## Author Response (AR2)

**Response to comments**

**Paper #:** essd-2022-289

**Title:** Lake Surface Temperature Dataset in the North Slave Region Retrieved from Landsat Satellite Series – 1984 to 2021

**Journal:** Earth System Science Data

**Reply to Reviewer #1**

The paper of Attiah et al. "Lake Surface Temperature Dataset in the North Slave Region Retrieved from Landsat Satellite Series - 1984 to 2021" brings an interesting overview of long-term dataset in the high latitude regions where the historic (but also recent) in situ observations are scarce. It analyses more than 500 lakes which also allow to analyse spatial variability of the temperature across the whole region.

I appreciate the detailed description of the methodology including data quality control. As not being a remote sensing expert I can't judge on every detail of the process. Therefore, I stress that this issue is hopefully reviewed by other reviewers.

We are thankful to the reviewer and appreciate their suggestions and valuable and positive comments for improving the manuscript. We have addressed or responded to all comments to improve the quality of this manuscript. Below, we provide the answers to the comments and questions raised by the reviewer. For convenience, comments from this Reviewer are provided in black text. Responses to each comment are provided in blue text and addition to manuscript is in purple text (font 10).

I would suggest including more details on the validation of the dataset against the in situ observations. This is to me essential for the reader/user as this basically can tell us something about how reliable the Landsat-derived results are. As I understand, the observations are made manually, but some of them with use of automatic loggers. You demonstrated that the individual datapoints are well represented, but are the statistical parameters derived from in situ and remotely sensed datasets comparable in terms of open water season average LST for example?

Thanks for the comment. As suggested,  LST values for a few statistical parameters have been calculated to further compare LST from in situ measurements with Landsat-derived values. The average yearly LST, total LST average and variance for available data are reported in Table 2 (see below). This analysis is added to the revised version of the manuscript from line 295.

Statistical parameters, including average yearly LST for open water season, total average LST and variance, were calculated for available ECCC and DataStream in-situ data and compared against North Slave LST, which is highlighted in Table 2. Absolute differences calculated for the statistical parameters ranged from 0.1°C to 1°C. The highest absolute difference for the average LST for open water between the two datasets was 1°C calculated for the year 2000 of the ECCC data. The variance was 0.6°C and 0.3°C for the ECCC and Datastream, respectively.

Differences between the total LST average were the lowest, with 0.3°C from 1999 to 2003 and 0.1°C from 2014 to 2019.

**Table 2: Comparison of yearly average LST, average LST and variance between North Slave LST and In-Situ LST.**

| Statistical Parameters | Data | Period | In-Situ LST (°C) | North Slave LST (°C) | Absolute Difference (°C) |
|---|---|---|---|---|---|
| Average LST for open water season (June – September) | ECCC | 1999 | 11.9 | 12.2 | 0.3 |
| | | 2000 | 10.8 | 11.8 | 1 |
| | | 2001 | 13.7 | 14.2 | 0.5 |
| | | 2002 | 11.1 | 11.2 | 0.1 |
| | | 2003 | 12.2 | 12.5 | 0.3 |
| | DataStream | 2014 | 13.7 | 14.2 | 0.5 |
| | | 2015 | 15.1 | 14.5 | 0.6 |
| | | 2016 | 16.1 | 16.4 | 0.3 |
| | | 2017 | 15 | 15.5 | 0.5 |
| | | 2019 | 14.2 | 13.3 | 0.9 |
| Total LST Average for open water season | ECCC | 1999-2003 | 12.3 | 12.6 | 0.3 |
| | DataStrean | 2014-2019 | 14.9 | 14.8 | 0.1 |
| Variance | ECCC | 1999-2003 | 15.5 | 14.9 | 0.6 |
| | DataStream | 2014-2019 | 3.8 | 4.1 | 0.3 |

I think it is essential to distinguish between water LST and temperature of a frozen/snow covered lake. I think it would be worth including this information in the dataset. This would also allow the user to estimate changes in lake ice cover duration, date of freezing and melting.

Thanks for the comment. As suggested by the reviewer, a few new parameters have been included in the dataset. First, the total number of LST pixels for ice/snow and for open water, respectively, for each lake, has been extracted for each given day. Furthermore, the percentage of ice pixels was estimated for each available day of all lakes in the dataset. With this, we believe users can determine their thresholds in deriving lake ice phenology from the data, including lake ice cover duration, date of freezing and melting (see description below).

You stated that the aim of your work is to distribute the data among the public/authorities. I don't think that any of these people would be able to process or use the data in its present form. I think that it would be great if you can also provide the reader/user with processed dataset of LST average values (for example) – in a table form. My suggestion would be to generate annual average LST for open water season, length of the open water, melt date, freezing date.

Thanks for the great suggestion. Processed tabular LST data has been generated for each lake studied and included in our dataset. Each lake name will be followed with longitude and latitude to allow for easy query based on location (e.g., AcastaLake_-115.564 _ 65.3783). This data includes the mean LST, median LST, maximum LST, minimum LST, total ice-covered pixels, total open-water pixels, percentage of ice pixels, area of Lake, lake area captured, Landsat sensor data and tile from which it was extracted. Additionally, another tabular LST data with monthly LST means for all lakes studied has also been included in the dataset.

The description of this dataset is included in the modified manuscript which shows the column structure and information contained in the table associated with the two new types of tabular data generated for all lakes. This has been added in from lines 375 in the manuscript.

**4.4.2 Tabular Data of LST**

The second type of data included in the North Slave LST dataset is the tabular data containing LST statistics on individual study lakes for a given day. Derived attributes include the minimum, maximum, median, mean, number of ice cover pixels, number of open-water pixels, percentage of lake captured and other lake properties. Table 4 below highlights the column/field names from the tabular data and what they represent. This tabular data is generated for each lake and is included in the dataset. Each filename consists of the lake name followed by longitude and latitude for easy query based on location (e.g., "AcastaLake_-115.564 _ 65.3783"). Additionally, monthly means were calculated for each lake and combined as one file in the dataset.

**Table 4: Columns names of the tabular dataset and the description.**

| Column | Description |
| --- | --- |
| Lake_Name | Name of the Lake from which the lake surface temperature was retrieved. The name of lakes was predominantly derived from the Water file-Lakes and Rivers polygons data from Statistics Canada. Lakes' unknown names were prefixed "NoNameLake" and a number |
| Date | The date which the lake surface temperature(LST) represents |
| Year | The year of the LST in the format "YYYY." |
| Month | The month of LST in the format "MM." |
| Day | Day of LST in the format "YY." |
| Maximum_Temperature | The maximum LST recorded on the lake at a given time in degrees Celcius (°C) |
| Minimum_Temperature | The maximum LST recorded on the lake at a given time in degrees Celcius (°C) |
| Median_Temperature | The median LST from all available pixels in degrees Celcius (°C) |
| Mean_Temperature | The mean LST from all available pixels degrees Celcius (°C) |
| Total_Pixels | Total number of pixels representing the lake |
| LST_Pixels | Number of pixels with LST values retrieved from the lake |
| Percentage_LST_Pixels | Total percentage of pixels with LST values captured from the lake. Values are rounded to the nearest 1 |
| Count_Water_Pixels | Number of LST pixels values greater than 0 retrieved from the lake at a given time |
| Count_Ice_Pixels | Number of LST pixels values less than 0 retrieved from the lake at a given time |
| Percentage_Ice_Pixels | Total percentage of ice pixels captured from the lake at a given time. Values are rounded to the nearest 1. |
| Landsat_Row_Path | Tile name, Row and path of the Landsat from which LST was retrieved |
| Lake_Area | Surface Area of the Lake in square kilometres ($km^2$) |
| HyLak_ID | The ID is derived from the HydroLAKES dataset. Lakes with no ID are indicated with 0. |
| HyLak_Depth | The average depth of the lake derived from the HydroLAKES dataset in meters (m) |
| HyLak_Volume | The volume of the lake derived from the HydroLAKES dataset is million cubic meters (1 mcm = $0.001km^3$) |
| HyLak_Elevation | Elevation of the lake surface derived from HydroLAKES dataset in meters above sea level |
| Long(m) | Longitude point on the lake in meters |

| Lat(m) | Latitude point on the lake in meters |
|---|---|
| Long(DD) | Longitude point on the lake in decimal degrees |
| Lat(DD) | Latitude point on the lake in decimal degrees |
| Monthly_Mean_Temperature | The mean LST on the lake for a given month |

In addition to the above major edits, minor edits were done to improve the manuscript.

**Reply to Reviewer #2**

Thank you for creating "Lake Surface Temperature Dataset in the North Slave Region Retrieved from Landsat Satellite Series". Without a doubt, it will be useful for both scientific and applied research in the country with the largest number of lakes on the planet. Its relevance increases in the conditions of global and especially regional (high latitudes) warming.

We are thankful to the reviewer, and we appreciate their suggestions and valuable and positive comments for improving the manuscript. We have addressed or responded to all comments to improve the quality of this manuscript. Below, we provide the answers to the comments and questions raised by the reviewer. For convenience, comments from this Reviewer are provided in black text. Responses to each comment are provided in blue text.

Firstly, this refers to the typification of lakes according to the origin of their bottoms. Despite quite probable differences in morphometry, lake bottoms of the same origin have common similar patterns in the distribution of depths and, consequently, temperature characteristics, including the surface temperature of lake waters. The genetic grouping of lakes will make it possible to better understand the features of the temperature regime of lakes and their changes in space and time.

Secondly, the temperature regime of lakes is largely influenced by whether they are transit for runoff or not (without outflow). Grouping lakes on this basis will make your database even more valuable for science and practice.

It is quite obvious that the temperature regime of lakes is influenced by a much larger set of factors. However, the above groupings will take a big step forward in effectively using the database you have created.

We appreciate the comment and acknowledge the necessity of grouping lakes based on genetic properties such as origin of lake bottoms, as well as lake inflow and outflow. Unfortunately, we do not have such information on the vast number of lakes in our study area. However, going along the lines of the recommendation, all other geographical properties and parameters we could access from other sources for lakes studied have been included in the modified dataset. The data was derived from the HydroLAKES data cited in our manuscript and included in the data description. A table with the summarized names and properties is included below and is included in the manuscript. We believe that with this added information, users of the dataset can set thresholds for which they choose to group lakes based on geographical and morphometrical properties. For

example, they can group lakes based on depth, area, volume and elevation, among others. Details of the added properties has been included in the manuscript from lines 375

**4.4.2 Tabular Data of LST**

The second type of data included in the North Slave LST dataset is the tabular data containing LST statistics on individual study lakes for a given day. Derived attributes include the minimum, maximum, median, mean, number of ice cover pixels, number of open-water pixels, percentage of lake captured and other lake properties. Table 4 below highlights the column/field names from the tabular data and what they represent. This tabular data is generated for each lake and is included in the dataset. Each filename consists of the lake name followed by longitude and latitude for easy query based on location (e.g., "AcastaLake_-115.564 _ 65.3783"). Additionally, monthly means were calculated for each lake and combined as one file in the dataset.

**Table 4: Columns names of the tabular dataset and the description.**

| Column | Description |
|---|---|
| Lake_Name | Name of the Lake from which the lake surface temperature was retrieved. The name of lakes was predominantly derived from the Water file-Lakes and Rivers polygons data from Statistics Canada. Lakes' unknown names were prefixed "NoNameLake" and a number |
| Date | The date which the lake surface temperature(LST) represents |
| Year | The year of the LST in the format "YYYY." |
| Month | The month of LST in the format "MM." |
| Day | Day of LST in the format "YY." |
| Maximum_Temperature | The maximum LST recorded on the lake at a given time in degrees Celcius (°C) |
| Minimum_Temperature | The maximum LST recorded on the lake at a given time in degrees Celcius (°C) |
| Median_Temperature | The median LST from all available pixels in degrees Celcius (°C) |
| Mean_Temperature | The mean LST from all available pixels degrees Celcius (°C) |
| Total_Pixels | Total number of pixels representing the lake |
| LST_Pixels | Number of pixels with LST values retrieved from the lake |
| Percentage_LST_Pixels | Total percentage of pixels with LST values captured from the lake. Values are rounded to the nearest 1 |
| Count_Water_Pixels | Number of LST pixels values greater than 0 retrieved from the lake at a given time |
| Count_Ice_Pixels | Number of LST pixels values less than 0 retrieved from the lake at a given time |
| Percentage_Ice_Pixels | Total percentage of ice pixels captured from the lake at a given time. Values are rounded to the nearest 1. |
| Landsat_Row_Path | Tile name, Row and path of the Landsat from which LST was retrieved |
| Lake_Area | Surface Area of the Lake in square kilometres ($km^2$) |
| HyLak_ID | The ID is derived from the HydroLAKES dataset. Lakes with no ID are indicated with 0. |
| HyLak_Depth | The average depth of the lake derived from the HydroLAKES dataset in meters (m) |
| HyLak_Volume | The volume of the lake derived from the HydroLAKES dataset is million cubic meters (1 mcm = $0.001km^3$) |
| HyLak_Elevation | Elevation of the lake surface derived from HydroLAKES dataset in meters above sea level |
| Long(m) | Longitude point on the lake in meters |
| Lat(m) | Latitude point on the lake in meters |

| Long(DD) | Longitude point on the lake in decimal degrees |
|---|---|
| Lat(DD) | Latitude point on the lake in decimal degrees |
| Monthly_Mean_Temperature | The mean LST on the lake for a given month |